# Mapping SP-C co-chaperone binding sites reveals molecular consequences of disease-causing mutations on protein maturation

Kristine F. R. Pobre-Piza[1], Melissa J. Mann[1], Ashley R. Flory[1] & Linda M. Hendershot [1✉]

BiP co-chaperones ERdj4, ERdj5, and GRP170 associate in cells with peptides predicted to be aggregation prone. Here, extending these findings to a full-length protein, we examine two Interstitial Lung Disease-associated mutants (ILD) of surfactant protein C (SP-C). The TANGO algorithm, which identifies sequences prone to formation of β strand aggregates, found three such regions in SP-C: the N-terminal transmembrane (TM) domain and two sites in the intermolecular chaperone BRICHOS domain. We show the ILD mutants disrupt di-sulfide bond formation in the BRICHOS domain and expose the aggregation-prone peptides leading to binding of ERdj4, ERdj5, and GRP170. The destabilized mutant BRICHOS domain fails to properly insert its TM region in the ER membrane, exposing part of the N-terminal TM domain site. Our studies with ILD-associated mutant proteins provide insights into the specificity of ERdj4, ERdj5, and GRP170, identify context-dependent differences in their binding, and reveal molecular consequences of disease-associated mutants on folding.

[1] Department of Tumor Cell Biology, St. Jude Children's Research Hospital, 38105 Memphis, TN, USA. ✉email: linda.hendershot@stjude.org

Secretory pathway proteins enter the endoplasmic reticulum (ER) lumen co-translationally in an unfolded state where they encounter molecular chaperones and folding enzymes. These resident ER proteins aid and monitor proper maturation of nascent proteins in a process called ER quality control (reviewed in refs. [1–3]). This is achieved by their ability to recognize and protect regions of the protein that will ultimately be buried upon correct folding. The failure of a protein to reach its native state results in continued exposure of chaperone recognition sites and signals its disposal through ER-associated degradation (ERAD) (reviewed in refs. [4,5]). Nearly one-third of the proteins encoded by the human genome enter the ER during their biosynthesis[6]. Thus, ER chaperones must recognize a vast number of sequence unrelated proteins, while at the same time distinguishing between unfolded and folded proteins. Insights into this dilemma were first obtained for the ER Hsp70 family member, BiP[7–9]. These studies revealed that peptides as short as seven amino acids provided optimal binding, and amino acids with aliphatic side chains were preferred. It was calculated that peptides of this sort would occur about every sixteen amino acids on the average protein[8], thus explaining BiP's ability to serve as a general chaperone.

Hsp70 family members are assisted by DnaJ co-factors and nucleotide exchange factors, some of which bind directly to unfolded proteins and transfer them to their Hsp70 partners (reviewed in refs. [10,11]). Binding preferences of DnaJ proteins from bacteria[12,13] and yeast[14] revealed they also preferred short, hydrophobic peptides. The emerging consensus was that general molecular chaperones, like Hsp70s and DnaJ proteins, recognized sequences without secondary structure that were enriched in hydrophobic amino acids. These sequences would be shielded upon folding, establishing the paradigm of how these chaperones could distinguish between nascent unfolded proteins and their mature native structures.

To understand how BiP and its co-chaperones interact with clients in cells, we developed an ex vivo peptide binding screen, which allowed chaperone interactions to be detected within the complex chemical and physical environment of the ER[15]. Constructs encoding overlapping peptides from an immunoglobulin light chain and heavy chain were integrated in a vector downstream of an ER-targeting sequence and Ig λ constant domain that allowed the peptides to be isolated. In keeping with in vitro screens[8,9], binding sites for BiP and ERdj3, which is structurally related to E. coli DnaJ and yeast Ydj1[16,17], occurred frequently throughout these two clients[15]. However, binding patterns for the ERdj4, ERdj5, and GRP170 co-chaperones, which had not been previously queried, were identical to each other, occurred more rarely than BiP or ERdj3 sites, and were readily eliminated or introduced by limited mutagenesis. The peptides were particularly rich in aromatic residues, which are generally less frequent in proteins of all species[18], likely due to their higher propensity to form aggregates[19]. In fact, the TANGO algorithm, which was developed to identify regions in proteins prone to β strand aggregate formation[20], readily identified all five co-chaperone binding sequences in these two clients.

In this study, we sought to determine if the TANGO algorithm could be used to predict binding sites for these BiP co-chaperones in the context of a full-length protein. We chose two disease-associated mutants of surfactant protein C (SP-C) that were reported to bind ERdj4 and ERdj5[21]. SP-C is synthesized by lung alveolar type 2 cells as a 197 amino acid proprotein that enters the ER co-translationally, folds, and assembles into non-covalent trimers. After passing ER quality control and being transported through the secretory pathway, SP-C undergoes multiple post-translational processing steps to produce a mature, palmitoylated peptide of 35 amino acids, which is incorporated into lamellar bodies and released by regulated exocytosis[22,23]. A significant portion of the proprotein encodes a BRICHOS domain, which acts as an intramolecular chaperone for integrating the unusual valine-rich TM domain. However, the mechanism by which the BRICHOS domain helps to integrate the TM in cells remains incompletely understood. In vitro studies performed with purified TM peptides and recombinant BRICHOS protein argue the chaperoning activity of this domain can occur in trans[24]. Numerous SP-C mutations have been identified that cause Interstitial Lung Disease (ILD), the majority of which map to the BRICHOS domain and result in the formation of amyloid-like aggregates[25–27]. ILD is an autosomal dominant disorder; however, the consequences of interactions between the wild-type and mutant protein have not been well-characterized in cells.

Here, we show that the TANGO algorithm identifies three sites in the SP-C sequence with varying aggregation propensity. Two are in the BRIOCHOS domain and the other is the transmembrane region, all of which should be inaccessible to the co-chaperones if SP-C matures properly. We find both ILD-associated mutants affect proper disulfide bond formation in the BRICHOS domain leading to exposure of the two potential co-chaperone sites in this domain, only one of which is recognized by the co-chaperones. Misfolding of this domain further impairs proper integration of the transmembrane domain, thus exposing a third site that the co-chaperones bind. Our studies to understand how the ILD-associated mutations lead to increased co-chaperone binding provide insights into normal SP-C biosynthesis and reveal how these mutations adversely affect normal maturation.

## Results

**All three co-chaperones bind to ILD-associated SP-C mutants**. If ERdj4, ERdj5, and GRP170 all recognize identical types of sequences, we surmised that proteins reported to bind one of these co-chaperones should bind the other two. Two disease-associated SP-C mutants, L188Q and one arising from deletion of exon 4 from the mRNA (Δ exon 4), which removes amino acids 109–145 from the BRICHOS domain, were examined for co-chaperone binding using immunoprecipitation-coupled western blotting analyses. In keeping with a previous report[21], both mutants showed increased binding to ERdj4 and ERdj5 compared to wild-type SP-C (Fig. 1). When their binding to GRP170 was examined, we found they also associated more strongly with GRP170 (Fig. 1).

To determine if this subset of co-chaperones bound aggregation-prone regions in another secretory pathway client, we analyzed the entire protein sequence of human SP-C with the TANGO algorithm, and three potential sequences were identified (Supplementary Fig. 1a, b). Site 1 is predicted to be highly aggregation-prone, but this region corresponds to the transmembrane segment, which if properly integrated into the ER membrane should not be detected by luminal ER chaperones. Sites 2 and 3 are in the BRICHOS domain, which serves as an intramolecular chaperone for the valine-rich transmembrane domain[24,28]. We examined their location on the BRICHOS domain ribbon structure[27], which revealed site 2 was comprised of three β strands, while site 3 was encoded by the C-terminal portion of a final α helix (Supplementary Fig. 1c). A space-filling model indicated these two sites should be largely buried when the BRICHOS domain is folded, together arguing that none of the three sites identified by the TANGO algorithm should be accessible on a well-folded and membrane-embedded SP-C proprotein.

**Identifying effects of ILD mutations on BRICHOS domain disulfide bonds**. The PyMOL visualization system was used to

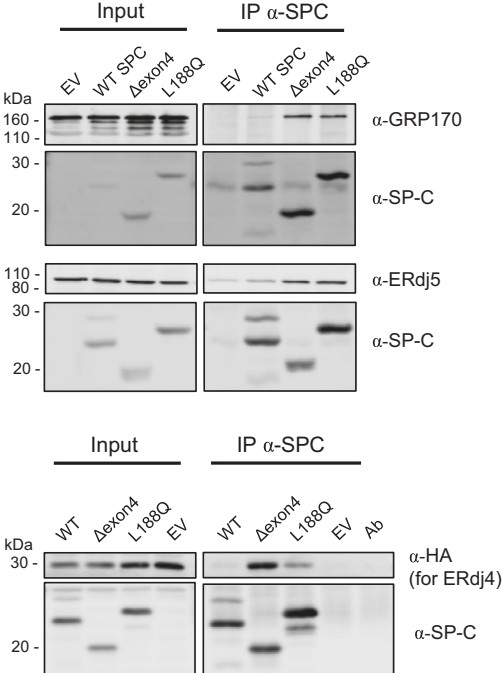

**Fig. 1 GRP170, ERdj4 and ERdj5 all bind SP-C mutant proteins.** Cells were transfected with cDNAs encoding the indicated co-chaperones along with wild-type SP-C and disease-associated mutants. The 3HA-tagged ERdj4 was used to distinguish it from SP-C because ERdj4 and SP-C have similar molecular weights. Lysates were prepared 24 h after transfection, and cell-equivalent amounts of lysates were subjected to immunoprecipitation coupled with western blot analyses. Anti-SP-C was used to isolate the clients, which were blotted with α-GRP170, α-HA (for ERdj4), α-ERdj5, or α-SP-C antibodies as indicated. For the SP-C/ERdj4 panel, an additional lane was included with the α-SP-C antibody (Ab), as a control for the signal from the precipitating antibody light chain, which has a similar mobility as ERdj4 Source data are provided as a Source Data file. Blots represent n = 3.

map the four cysteines on the crystal structure of the BRICHOS domain (PDB: 2YAD)[27], revealing the presence of a disulfide bond between C120 and C148 (pink) and between C121 and C189 (dark blue) (Fig. 2a). To determine if mutation of L188 (magenta) affected oxidative folding of SP-C, we analyzed disulfide bond formation in wild-type SP-C and the L188Q mutant. Cells expressing these proteins were radiolabeled, and the various SP-C proteins were isolated by immunoprecipitation. Electrophoresis under non-reducing conditions revealed two bands for the wild-type protein (Fig. 2b); both of which migrated slightly faster compared to their mobility when the sample was reduced (Supplementary Fig. 2b). This is indicative of the presence of longer range intramolecular disulfide bonds causing the protein to maintain a more compact form as it was separated by SDS-PAGE[29]. Examination of the L188Q mutant under non-reducing conditions revealed that the majority of this protein (~90%) was present as intermolecular, disulfide-linked oligomeric species, many of which were so large they remained at the top of the gel (Fig. 2b). Two less abundant, more quickly migrating species were also detected. One of these comprising ~6% of the protein co-migrated with the fully reduced protein (green square), and the other present as only ~4% of the protein possessed a single disulfide bond (red square) (Supplementary Fig. 2b), revealing the L188Q substitution leads to dramatic oxidative misfolding of the BRICHOS domain. Although C148 occurs earlier in the sequence and therefore would enter the ER lumen well before C189, it was noteworthy that there was little evidence of the C120–C148 bond

forming in the L188Q mutant. To more carefully examine the order of disulfide bond formation in SP-C, we characterized oxidative folding of mutants that disrupted each of the two disulfide bonds independently. To test the importance of the C120–C148 bond in BRICHOS domain folding, we engineered a serine substitution at C148, although no mutations of either cysteine in this bond have been identified in ILD patients. Examination of the C148S mutant under non-reducing conditions revealed that although ~40% of this mutant protein formed intermolecular disulfides, very few were very large species remaining at the top of the gel. Additionally, the 60% of the protein that remained monomeric formed an intramolecular disulfide bond that likely represents the C121–C189 bond (Supplementary Fig. 2b). Examination of the C189S construct demonstrated a pattern very similar to the L188Q mutant consistent with mutations at L188 affecting the formation of the C121–C189 bond. Over 80% of this mutant was present as large disulfide-linked oligomers with very little evidence of any monomeric protein possessing a disulfide bond. The differences between the oxidative folding of the C148S, and C189S mutants argues that formation of the C120–C148 bond is completely dependent on the formation of the C121–C189 bond and likely occurs last, as the disruption of this bond has less of an effect on the formation of the other bond even though C148 enters the ER well before C189. When examining the Δ exon 4 mutant, which includes C120 and C121 thereby disrupting both disulfide bonds, we found nearly 60% of this mutant formed intermolecular disulfide bonds (Fig. 2b), and the remainder possessed a non-native bond (Supplementary Fig. 2b), indicating that this disease-associated mutant also has dire consequences for folding of the BRICHOS domain. When the same samples were analyzed under reducing conditions, two molecular species were detected for wild-type SP-C, which corresponded to the unmodified (white triangle) and palmitoylated forms (blue triangle), respectively[22] (Fig. 2c). While 28.7% of the wild-type protein present during a 3 hr label was palmitoylated, only 7.8% of the C148S mutant was. Similar analysis of the L188Q, C189S, and Δ exon 4 proteins under reducing condition identified a single predominant band that represented the unmodified protein, since these mutants fail to pass ER quality control and as a result are not trafficked to the Golgi where palmitoylation occurs[22,30]. It is noteworthy that a portion of the large intermolecular bonded species in the L188Q mutant remained resistant to reducing agents, although this was not the case for the C189S and Δ exon 4 mutant proteins. We surmise this may be due to the presence of four unoxidized cysteines in L188Q, versus 3 and 2, respectively, in the other two mutants, resulting in more complex bond formations in L188Q.

**Co-chaperone binding to TANGO-identified BRICHOS domain peptides.** We reasoned that the misfolding of the BRICHOS domain caused by the mutation of L188, might expose Sites 2 and 3 allowing the binding of co-chaperones to the SP-C mutant proteins in the ER lumen. To determine if either of these were co-chaperone binding sites, we first engineered them into the ER-Cλ peptide expression vector used in the in-cell peptide-binding screen[15]. Briefly, the ER-Cλ vector encodes an ER-targeting signal sequence engineered up-stream of an Ig λ constant region (Cλ) domain, which was followed by a 21 amino acid glycine/serine repeat and ending with either an NVT or QVT motif. The former was used to test for translocation into the ER lumen where chaperones could be encountered, and the latter was used to ensure the BiP/co-chaperone system was queried instead of the lectin chaperones. The entire 26 amino acid site 2 was introduced into this construct. Site 3 has only 6 amino acids predicted to be aggregation prone, so we engineered this site

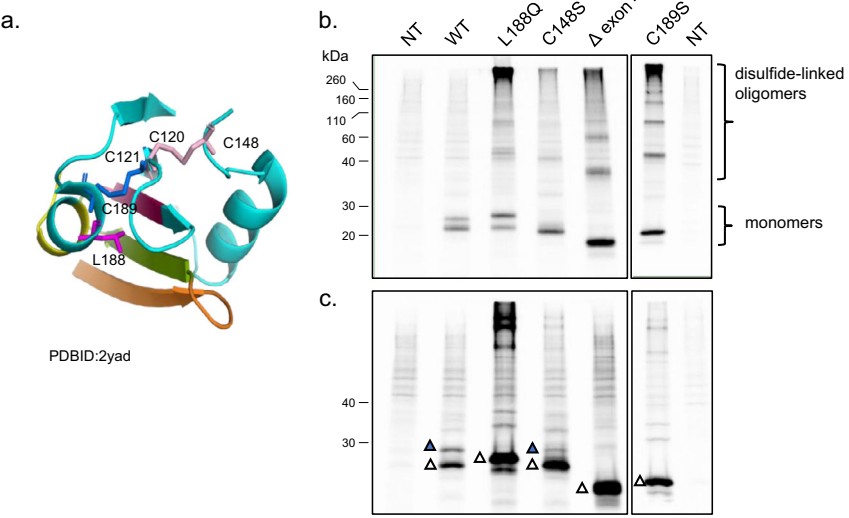

**Fig. 2 ILD-associated mutations prevent the formation of native intra-domain disulfide bonds. a** A ribbon depiction of the BRICHOS domain was generated using the PyMOL visualization system with the intramolecular disulfide bonds shown and the position of L188 indicated. **b** Cells were transfected with cDNAs encoding either SP-C WT, the ILD-associated mutants L188Q and Δ exon 4, the C148S and C189S mutants designed to disrupt native disulfide bonds or left non-transfected (NT). Twenty-four hours after transfection, cells were metabolically labeled for 3 h, lysates were immunoprecipitated with anti-SP-C, and samples were electrophoresed under non-reducing conditions. Signals for the disulfide-linked oligomers and each of the monomeric bands were quantified by phosphorimaging, and the background signal from the non-transfected cells was subtracted before calculating the percent of each species ($n = 4$ independent biological replicates). **c** An aliquot of each of the same samples was reduced prior to SDS-PAGE analyses. The mobilities of unmodified (white triangle) and palmitoylated (blue triangle) SP-C proteins are indicated ($n = 4$ independent experiments). Source data are provided as a Source Data file.

along with 19 flanking amino acids into the expression vector (schematics shown in Supplementary Fig. 3). We also made peptide constructs with mutations in both site 2 and 3 that were no longer indicated to be aggregation prone (Supplementary Fig. 3a, b, d, e). In the case of site 2, a single amino acid was altered to alanine on each of the three β strands comprising the site, whereas a single F → A substitution was made in site 3 (Supplementary Fig. 3a, d). Since site 3 containing peptide included L188, we also made constructs with glutamine at this position. However, this did not significantly alter the TANGO prediction (Supplementary Fig. 3d). Analysis of the NVT versions of these constructs revealed that both wild-type and mutated versions of both sites were glycosylated (Supplementary Fig. 3c, f), demonstrating that each of them was fully translocated into the ER lumen where they could encounter the ER co-chaperones. To examine co-chaperone binding, QVT versions of the various peptide constructs were used. Using immunoprecipitation-coupled western blotting analyses, we readily detected association of GRP170, ERdj5, and ERdj4 with the construct that included the site 2 peptide, whereas mutation of this sequence to one predicted not to be aggregation-prone disrupted binding to all three of them (Fig. 3a). Conversely, we were unable to detect binding of any of the co-chaperones to the peptide-containing site 3 (Fig. 3b).

**Contribution of BRICHOS domain sites on co-chaperone binding to SP-C.** To determine if either site 2 or site 3 was responsible for co-chaperone binding to the disease-associated SP-C proteins, we engineered these mutations onto the full-length L188Q protein both individually and together. The resulting L188Q constructs were tested for co-chaperone binding using immunoprecipitation-coupled western blot analyses (Fig. 4a). Due to the differences in expression levels, co-chaperone binding was normalized to the amount of the various L188Q proteins expressed as described in the methods section. Engineering the same substitutions in site 2 on the full-length L188Q protein,

which blocked co-chaperone association with the correspondingly mutated peptide, reduced GRP170 binding to ~30% of parental L188Q protein (Fig. 4b). Conversely, the site 3 mutation on full-length L188Q had no significant effect on co-chaperone binding either alone or when combined with the site 2 mutations (Supplementary Fig. 4). The effect of the site 2 mutations on ERdj4 binding to the full-length L188Q protein was similar to that observed for GRP170, and inclusion of the site 3 mutation together with those of site 2 had a very little additional effect on its binding. Although mutation of site 2 in the L188Q protein also significantly reduced binding of ERdj5, the effect was less dramatic and statistically different than that observed for GRP170 or ERdj4 binding (Fig. 4b, Supplementary Fig. 4). Thus, unlike data obtained from the peptides where mutations affected co-chaperone binding equally, mutation of site 2 on the full-length L188Q protein reduced binding of the three co-chaperones to different extents suggesting context-dependent effects in their binding.

**A portion of the SP-C L188Q transmembrane extends into the ER lumen.** Although mutation of site 2 led to a significant reduction of co-chaperone binding to the L188Q protein, a readily detectable amount of binding remained for all three of them. This led us to conclude that either there was another co-chaperone binding site present in this SP-C mutant that was not detected by the TANGO algorithm, or site 1, which corresponds to the transmembrane domain, was not completely integrated into the ER membrane when the BRICHOS domain structure was disrupted, providing an additional co-chaperone binding site. To examine the latter possibility, we took advantage of the fact that asparagine residues present in the consensus motif, N-X-S/T, must be ~12 amino acids inside the ER lumen to reach the active site of the membrane-tethered oligosaccharyl transferase (OST) complex for co-translational glycosylation[31]. A single N-linked glycosylation site was introduced at defined points in the wild-type and L188Q mutant proteins, which are both normally non-

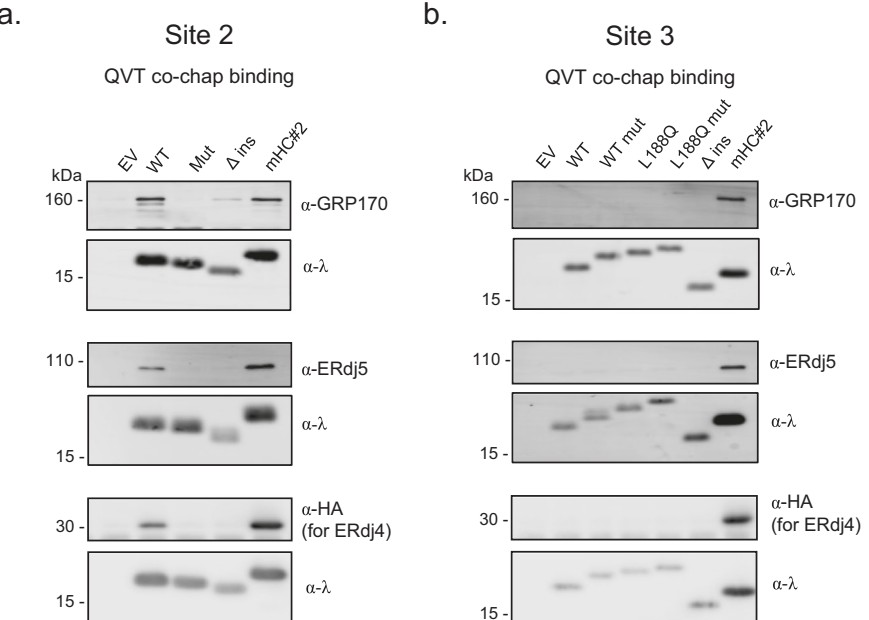

**Fig. 3 Co-chaperones bind to aggregation-prone site 2 but not to site 3.** Site 2 (**a**) and site 3 (**b**) peptides indicated by TANGO were inserted into the ER-Cλ-QVT construct and co-expressed with either GRP170, ERdj5, or ERdj4-3HA. Peptide-containing constructs were isolated with anti-λ, separated by reducing SDS-PAGE, and transferred for blotting with the indicated antibodies. A construct with no peptide insert (Δ ins) was used as a negative control, and a construct (mHC#2) from original study was used as a positive control. Sequence of mutants and TANGO plots are shown in Supplementary Fig. 3. Since site 3 included L188, which is mutated in ILD, constructs were also made with glutamine (Q188) at this position. Source data are provided as a Source Data file. Representative images for $n = 3$.

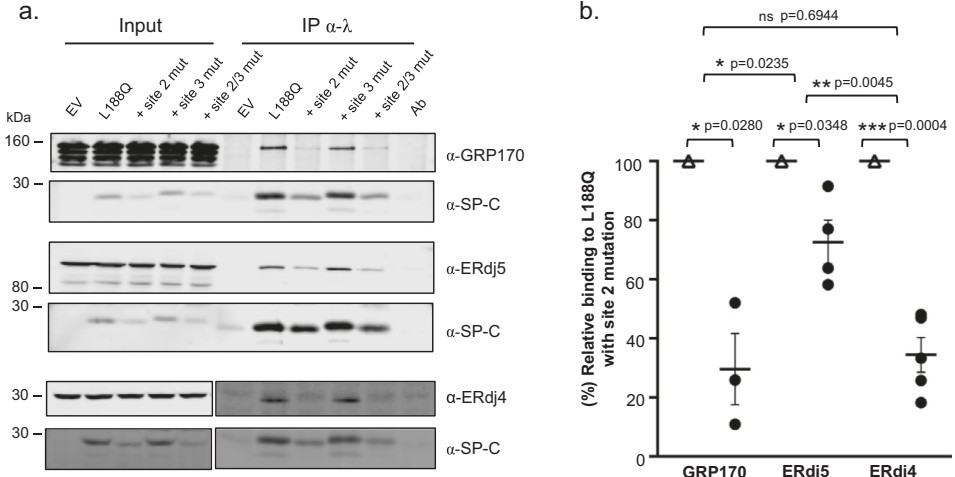

**Fig. 4 Mutation of site 2 in the L188Q mutant reduces co-chaperone binding, but significant binding remains. a** Cells were co-transfected with cDNAs encoding L188Q (parental), the indicated L188Q mutants in which site 2 and site 3 mutations had been engineered, along with either GRP170, ERdj5, or ERdj4-3HA. Twenty-four hours after transfection, cells were harvested and lysed with the appropriate buffers as described in the methods section. 1.5% of the lysates were retained for input, and the remaining cell lysates were immunoprecipitated with anti-SP-C. Samples were separated by reducing SDS-PAGE, and both SP-C and associated co-chaperones were detected by western blotting with specific antibodies. **b** Signals for binding of each co-chaperone to the site 2 mutant were first normalized to mutant L188Q expression levels compared to that of parental L188Q. The binding of each co-chaperone to the mutant constructs (circle) was expressed as a percent of that bound to the parental construct (triangle). For GRP170 binding, $n = 3$ independent experiments, for ERdj5, $n = 4$ independent experiments, and for ERdj4, $n = 5$ independent experiments. Standard deviation (SD) and standard error of the mean (SEM) were determined. Horizontal lines and error bars represent the mean and (SEM), respectively. The difference in co-chaperone binding between SP-C L188Q (parental) and each SP-C mutant (SP-C L188Q site 2, site 3, and site 2/3) was assessed using one-sample $t$ test. Differences between the binding of co-chaperones (GRP170 vs ERdj5, ERdj5 vs ERdj4, and GRP170 vs ERdj4) to each SP-C mutant were estimated using unpaired $t$ test. A threshold $p < 0.05$ was statistically significant. Source data are provided as a Source Data file.

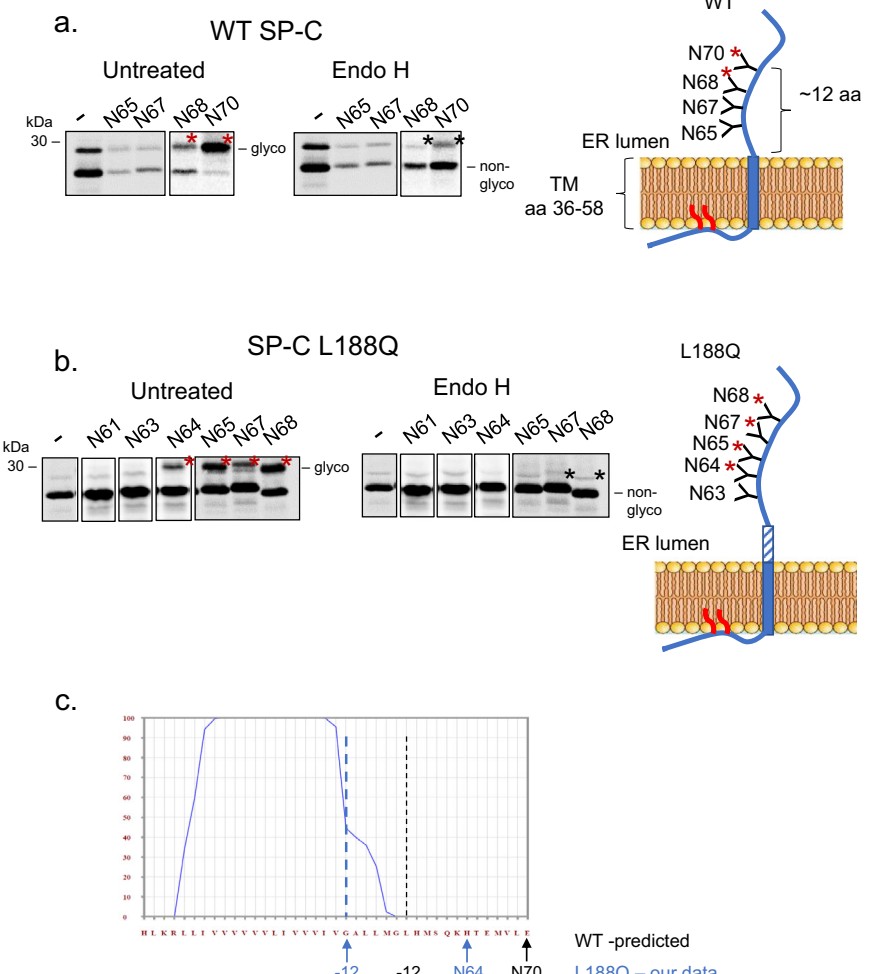

**Fig. 5 The transmembrane domain of the SP-C L188Q mutant extends further into the ER lumen than the wild-type SP-C protein.** Wild-type (**a**) and L188Q mutant (**b**) constructs were engineered with glycan acceptor sites at the indicated residues, which were used to transfect cells. After 24 h, cells were metabolically labeled for 30 min, and SP-C proteins were immunoprecipitated with α-SP-C. Isolated proteins were either left untreated or incubated with EndoH to remove glycans, and then separated by SDS-PAGE. The glycosylated forms are indicated with a red asterisk and the palmitoylated form, which co-migrates with the glycosylated form, is identified with a black asterisk. Schematic shows positions of engineered glycan acceptor sites. Those sites that are modified are indicated with a red asterisk (representative images for $n = 3$ replicates). **c** TANGO plot for site 1, and the membrane boundary for wild-type SP-C is shown with a black dotted line and the predicted boundary for L188Q with a blue dotted line. Source data are provided as a Source Data file.

glycosylated proteins. This technique was used previously to identify the membrane boundary of the wild-type SP-C protein using an in vitro translation/translocation assay[32]. When the radiolabeled wild-type glyco-constructs were isolated, we detected an increase in the intensity of the slower migrating band in the N68 construct compared to the non-glycan engineered wild-type protein and a more dramatic increase of this species in the N70 protein. Treatment of these samples with Endoglycosidase H (Endo H) revealed that the increase in this band represented N-linked glycosylation that co-migrated with the palmitoylated form. The glycan acceptor site engineered at N70 is precisely 12 amino acids from the predicted boundary of the transmembrane and luminal portions of the SP-C protein (Fig. 5c, black dotted line) and is in keeping with data obtained previously for wild-type SP-C[32]. When the radiolabeled SP-C L188Q constructs were similarly examined, a significant portion of the N64 mutant protein was glycosylated, as were the N65, N67, and N68 constructs (Fig. 5b). This revealed that a portion of the transmembrane domain predicted by the TANGO algorithm to be prone to β-aggregate formation now entered the ER lumen when

BRICHOS domain folding was disrupted by the L188Q mutation (Fig. 5c). A glycan acceptor site engineered very near the N-terminus was not modified, indicating the entire transmembrane domain did not fully enter the ER lumen (Supplementary Fig. 5a).

To assess if the portion of the TM domain that entered the ER might interact with the co-chaperones and contribute to the overall binding signal observed for the L188Q mutant protein, we inserted this segment into the NVT and QVT ER-Cλ constructs along with two mutations in this region that were predicted to reduce aggregation propensity. Several additional amino acids preceding this sequence were included to increase the length of the site (Supplementary Fig. 5a). All three of the constructs entered the ER as determined by analyzing the NVT constructs (Supplementary Fig. 5c). Binding of all three co-chaperones to the partial site 1 sequence could be detected, however, the TANGO-guided mutations only partially reduced this binding (Supplementary Fig. 5c). This might suggest either that the TM region extends further into the ER lumen than the glycosylation data indicates, the partial decrease in binding observed with the

peptide construct was not sufficient to detect a change on the full-length protein, or that an additional co-chaperone binding site exists that was not identified by TANGO. Despite a concerted effort, we were unsuccessful in distinguishing between these possibilities.

**Mutant SP-C blocks transport of the wild-type protein to the Golgi.** Due to the autosomal dominant nature of ILD, we tested the effects of co-expressing mutant SP-C on the maturation of the wild-type protein, as a number of informative mutants had been generated. We employed a 30-minute pulse-labeling to focus on early events in biosynthesis, and first examined the effect of mutant SP-C co-expression on the membrane integration of the wild-type protein using our construct with a glycan engineered at N67. In keeping with the data in Fig. 5, only 5.6% of this protein was glycosylated when expressed alone, indicating that over 90% of the protein was properly integrated into the ER membrane. Co-expression of the L188Q mutant, which has a misfolded BRICHOS domain, did not increase the amount of glycosylation of the engineered wild-type protein, demonstrating mutant SP-C co-expression did not adversely affect normal integration of the valine-rich wild-type TM domain (Supplementary Fig. 6a).

To examine the effects of mutant SP-C expression on oxidative folding of the wild-type SP-C protein and its transport to the Golgi, the L188Q mutant was modified with a 3X-HA peptide tag at its N-terminus, allowing it to be readily distinguished from both the unmodified and palmitoylated forms of the wild-type protein. This also provided a method to detect the association of mutant SP-C with the untagged wild-type protein, which was not isolated with the anti-HA monoclonal (Supplementary Fig. 6b). Cells expressing the indicated constructs were metabolically labeled for 30 min, lysates were immunoprecipitated with the indicated antibodies, and isolated material was split for electrophoresis under either reducing or non-reducing conditions. Electrophoresis of the immunoprecipitated SP-C proteins under reducing conditions revealed when wild-type SP-C was expressed alone 17.6% was palmitoylated, whereas palmitoylation was reduced to 10% when it was co-expressed with the L188Q mutant (Fig. 6a). Isolation of L188Q mutant with the anti-HA

monoclonal co-precipitated a portion of wild-type untagged SP-C, and this pool of wild-type SP-C was almost entirely devoid of palmitoylation (Fig. 6a). Together this indicated that mutant SP-C prevented egress of wild-type SP-C, even though wild-type SP-C had properly integrated into the ER membrane. Electrophoresis of the other half of the sample under non-reducing conditions revealed co-expression of L188Q had a very minor effect on the oxidative folding of wild-type SP-C when the percent of disulfide-linked oligomers was corrected for the contribution of the wild-type protein to the total signal, which was calculated from values observed under reducing conditions. The shorter labeling time resulted in the L188Q protein existing as a broader smear of disulfide-bonded oligomers with a much larger pool of reduced and partially oxidized monomers (Fig. 6b), compared to it being present primarily as very large disulfide-linked oligomers when a 3 h labeling time was used (Fig. 2b). This suggests a progressive formation of increasingly larger disulfide-linked oligomers of L188Q.

## Discussion
We previously developed a in vivo peptide screen covering an immunoglobulin light chain and the two domains of an Ig heavy chain that are the focus of ERQC to identify binding preferences for BiP and 4 of its co-factors within the chemically complex environment of the ER[15]. That screen revealed that GRP170, ERdj4, and ERdj5 bound identical peptides that were rarer than those that BiP and ERdj3 bound. All 5 peptides recognized by the co-chaperones could be identified by the TANGO algorithm, which was developed to predict regions in proteins prone to form β strand aggregates[20]. In the present study, we explored the binding of these three co-chaperones in the context of a full-length protein, but first determined if the three sequences in SP-C identified by TANGO bound as peptides. We found a number of consistencies with the previous study and some clear deviations. In keeping with the previous study, Site 2 bound all three co-chaperones when introduced into the peptide expression vector, and mutations that reduced the TANGO prediction for aggregation inhibited binding of all three co-chaperones equally. Site 3 was comprised of only 6 amino acids with a low predicted

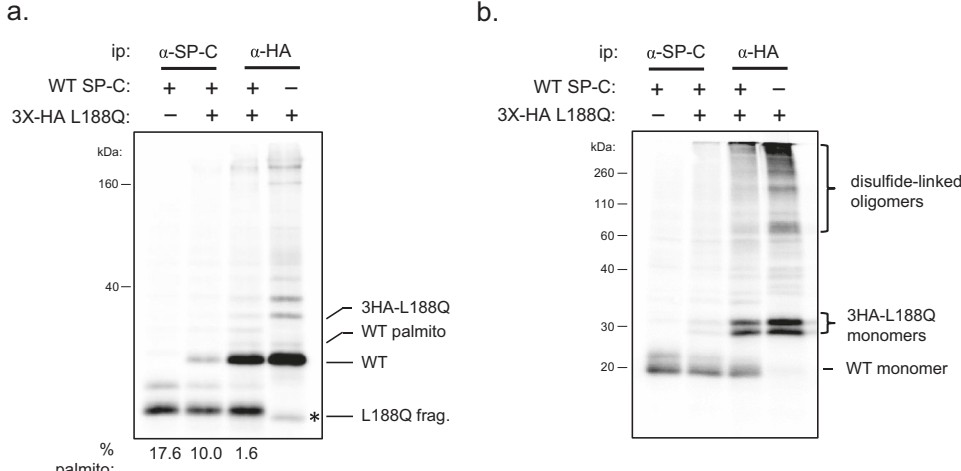

**Fig. 6 Co-expression of a SP-C disease-associated mutant adversely affects proper maturation of the wild-type SP-C protein. a** Cells were transfected with cDNA encoding untagged wild-type SP-C, 3HA-tagged L188Q, or a combination of the two constructs. After 24 h, cells were pulse-labeled for 30 min, lysates were prepared, and proteins were isolated with the indicated antibodies. The immunoprecipitated material was analyzed by reducing SDS-PAGE, and immunoprecipitation controls are shown in Supplementary Fig. 6b. The percent palmitoylation of the WT protein in each case was determined and is indicated below the lanes. The asterisk identifies a proteolytic fragment of L188Q that migrates slightly faster than the unmodified form of wild-type SP-C ($n = 5$ independent experiments). **b** The other half of samples analyzed in (**a**) were electrophoresed under non-reducing conditions ($n = 3$ independent experiments). Source data are provided as a Source Data file.

aggregation propensity, and 2 of the 6 amino acids were alanines, which disrupted co-chaperone binding when introduced into peptides[15] and full-length SP-C in this study. When this peptide was tested, we were unable to detect binding of any of the three co-chaperones. In our previous study, four of the five peptides were much longer than the aggregation prone region of Site 3, and even reducing their size from 25 to 12 amino acids generally had a significant effect on co-chaperone binding, arguing that unlike BiP larger sequences were required for recognition by the co-chaperones. The aggregation-prone region of one peptide in the previous study was only 7 amino acids in length, but it had a much higher aggregation propensity[15]. The lack of co-chaperone binding to site 3 reveals limits on the combination of size, amino acid composition, and aggregation propensity. Site 1 had the clearest distinction in the ability of the TANGO algorithm to predict co-chaperone binding. Although the aggregation-prone sequence we inserted into the peptide expression vector was relatively short (10 aa), the predicted propensity for aggregation was quite high. All three co-chaperones bound to this peptide in cells, but proline substitutions that dramatically reduced TANGO-predicted aggregation propensity had only a moderate effect on the binding of all three co-chaperones. Thus, our study revealed the ability of the TANGO algorithm to predict co-chaperone binding or loss of binding was not absolute.

Unlike the data obtained from peptide binding studies, when the site 2 peptide mutations were engineered into the full-length L188Q mutant protein, differences in the effects of the mutations on individual co-chaperone binding were revealed. The binding of GRP170 and ERdj4 was reduced by ~70% indicating this sequence was likely the major interaction site on SP-C for both of these co-chaperones. However, the binding of ERdj5 was only reduced by ~30% with these same mutations. It is noteworthy that ERdj5 is also a member of the protein disulfide isomerase family and has reductase activity[33]. We found the L188Q mutant was largely present in large disulfide-linked oligomers, which likely provided a distinct type of binding opportunities for this co-chaperone. A caveat to the data obtained with the site 2 mutations on the full-length L188Q protein, is that the three β sheets that comprise site 2 have been proposed to contribute to the chaperoning activity for the valine-rich TM region[27]. We were unable to identify amino acids to change the aggregation propensity that were not suggested to contribute to chaperone activity. Thus, our mutations introduced in site 2 in the context of the full-length protein may have resulted in a greater effect on site 1 egress into the ER.

It is noteworthy that all three of these co-chaperones have been reported to function in the identification of proteins for degradation instead of aiding the folding of BiP client proteins[11]. Our previous study revealed that these aggregation-prone sites are well-tolerated in the $V_H$ and $C_L$ domains, which fold stably in the clients queried, but their introduction in a domain that is unable to fold ($C_H$1) leads to disulfide-linked oligomers and the formation of NP40-insoluble aggregates[15]. This likely explains their tolerance in the BRICHOS domain of wild-type SP-C, the fact that ERdj4 and ERdj5 bind preferentially to the ILD-associated mutants, and reports that over-expression of both co-chaperones accelerates mutant SP-C degradation[21]. Aggregates are notoriously difficult for cells to clear, as metazoans do not have homologs of Hsp104 disaggregases, which perform this function in bacteria[34] and yeast[35]. A fairly recent study demonstrated that an interaction between Hsp70, Hsp110, and an Hsp40/DnaJ protein can disassemble protein aggregates in vitro[36]. GRP170 is the ER member of the Hsp110 family, which possess ATPase activity in addition to binding peptides, and the few peptides shown to bind yeast Hsp110 are rich in aromatic residues[37,38]. Thus, providing a possible link between

this aggregate-breaking complex and the types of sequences that our co-chaperones bind.

While determining if the various TANGO-predicted sequences were accessible to the co-chaperones, our studies had the additional benefit of providing insights on disease-associated SP-C mutants and the function of the BRICHOS domain in SP-C biogenesis, which has been the subject of a large body of elegant biophysical studies. The BRICHOS domain is stabilized by two intramolecular disulfide bonds that form between C120 and C148 and between C121 and C189, which provide resistance to urea and heat denaturation in vitro[39]. The ILD-associated L188Q mutation had a strong negative impact on the formation of the C121 and C189 bond, which secondarily affected the formation of the native C120 and C148 bond, resulting in large disulfide-linked oligomers that involved non-native bonds. L188 is the target of two other ILD mutations, and a total of four mutations disrupt this disulfide by altering either C121 or C189. Our data suggest that normally the C121-C189 bond forms first and is more critical to the stability of the BRICHOS domain, as only about ~4% of the L188Q mutant formed the second disulfide bond. Conversely, mutation of C148 still permitted formation of the C121 and C189 bond in ~60% of the mutant molecules, and particularly large disulfide-linked oligomers were not formed. In keeping with this finding, neither C120 nor C148 mutations have been identified in ILD, and this bond is not conserved in other proteins containing BRICHOS domains or even the BRICHOS domains of SP-C from all species[40].

Although it is well-established that SP-C encodes its own dedicated chaperone, it is less clear if the BRICHOS domain of each molecule aids in the insertion of its own TM domain or if the BRICHOS domain acts in trans to insert the TM of another nascent SP-C molecule. The most common ILD-associated mutation, I73T[26], occurs in the linker region between the TM and BRICHOS domains, which could be compatible with either a cis or trans activity, and like the BRICHOS domain, the linker is a frequent target of ILD mutations[41]. The possibility of a trans chaperone activity are supported by data showing degradation of the L188Q mutant is reduced by co-expression of wild-type SP-C[24], and the ability of a recombinant wild-type BRICHOS domain to bind to phospholipid membranes in which a poly-valine peptide was inserted[28]. In the present study, we demonstrated that the disulfide bond between C121 and C189 is critical for oxidative folding of the BRICHOS domain. The fact that C189 is only 8 amino acids from the C-terminus of the protein strongly argues against a cis chaperone activity unless the entire protein remains in the translocon during its biosynthesis.

ILD is an autosomal recessive disorder, begging the question as to whether this disease arises due to haplo-insufficiency or if the mutant protein also adversely affects the maturation of wild-type SP-C. Therefore, we tested the effects of an epitope-tagged SP-C mutant on the membrane integration and oxidative folding of wild-type SP-C, as well as its modification by palmitoylation, as an indication of it passing ER quality control and trafficking to the Golgi. We found that the non-native BRICHOS domain of L188Q was capable of engaging or being engaged by the wild-type SP-C monomers, but this did not significantly affect membrane integration or oxidative folding of the wild-type protein. This suggests the misfolded BRICHOS domain does not act as a dominant negative mutant in the biosynthesis of the wild-type protein. However, the association of the mutant protein prevented transport of the wild-type protein to the Golgi where SP-C is modified by the attachment of palmitic acid to two cysteines near the N-terminus. Previous studies revealed that homomeric assembly of SP-C via its transmembrane domain is required for its transport out of the ER[42], and that aggregation prone mutants like Δ exon 4 can trap the wild-type protein[43]. Our data argue

that the incompletely integrated TM of the L188Q mutant is still capable of interacting with the well-integrated TM of the wild-type protein. The chaperone-mediated retention of the mutant SP-C in turn traps properly matured wild-type subunits, and indirectly they prevent transport of normally matured SP-C molecules to the Golgi.

In summary, three BiP co-chaperones, ERdj4, ERdj5, and GRP170, were shown to bind identical peptides in cells, which were generated from antibody heavy and light chains, that were predicted to be aggregation prone by the TANGO algorithm. The present study was designed to extend these findings to an unrelated protein and to determine if TANGO-identified peptides served as co-chaperone binding sites on a full-length protein in cells. Our studies found that these co-chaperones also bound identical peptide sequences in SP-C, while revealing further insights/limitations in the ability of TANGO to predict their binding. Furthermore, the effects of mutations in these sequences in the context of full-length proteins were not identical, highlighting context-dependent factors in binding and revealing unique binding properties for the PDI family member ERdj5. The constructs made to identify co-chaperone binding in full-length SP-C mutants provided us with an opportunity to contribute strong support for the hypothesis that the BRICHOS domain acts in *trans* to chaperone the integration of SP-C's unusual transmembrane region into the ER bilayer and revealed insights into folding defects for ILD-associated SP-C mutants and the autosomal dominant nature of this protein folding disease.

## Methods

**Expression vectors, cloning, and mutagenesis.** The expression vectors used for chaperones and co-chaperones have been described previously: pcDNA3.1-GRP170 (human)[44], pCMV-ERdj5 (mouse)[45], pMT-hamster BiP[46], and pSG5-ERdj4-3HA with 3 HA tags at the C-terminus (mouse)[45], which allowed us to distinguish between SPC WT, L188Q, and ERdj4 which have similar molecular weights. The wild-type and mutant pro-surfactant protein C constructs pIRES2-EGFP-SP-C wild type (WT), pIRES2-EGFP-L188Q, and pIRES2-EGFP-delta exon 4 (Δ exon 4) were a generous gift from Dr. Timothy Weaver (Cincinnati Children's Hospital Medical Center)[47]. The expression vectors with SP-C peptide inserts were created using the ex vivo peptide expression vector (pSVL Δ insert) described previously[45]. Briefly, a 25-26-amino acid peptide sequence from SP-C was inserted in the flexible Gly/Ser (GS) linker, which is positioned after an ER-targeted λ light chain (LC) constant domain ($C_L$) and followed by a C-terminal NVT glycosylation site that is used to monitor translocation of the recombinant protein into the ER lumen. A corresponding QVT version of the various constructs was used to detect co-chaperone binding. Point mutations were engineered using the Q5 site-directed mutagenesis kit (New England Biolabs, NEB, E0554S). Primers were designed using the NEBaseChanger tool and are listed in Supplementary Table 1. Other point mutants (site 2 and site 3 mut) were made using Dpn-I PCR site-directed mutagenesis. The sequences of all DNA constructs were verified.

**Cell culture, transfection, and lysis.** 293 T cells were maintained at 37 °C and 5% $CO_2$ in Dulbecco's Modified Eagle's Medium (DMEM) (Corning, 15013-CV) supplemented with 10% (v/v) fetal bovine serum (FBS, Biotechne, S10350), 1% Antibiotic-antimycotic (Corning, 30-004-CI), and 1% L-glutamine (Corning, 25-005-CI). Only mycoplasma-free cells were used. Cells were plated 24 h before transfecting with the GeneCellin transfection reagent (BioCellChallenge, GC5000) according to the manufacturer's instructions. To detect co-chaperone binding, cells were co-transfected with cDNAs to express the indicated client and co-chaperone. Cell pellets were disrupted using NP-40 lysis buffer (0.5 M Tris-HCl pH 7.5, 0.15 M NaCl, 0.5% DOC, 0.5% NP40 substitute, 0.002% sodium azide) supplemented with 0.1 mM PMSF and 1x Roche complete protease inhibitor tablets and clarified by centrifugation for 15 min at maximum speed in a refrigerated microfuge. In the case of ERdj5 expression, cells were washed twice with PBS and lysed with NP-40 lysis buffer both of which contained 20 mM N-ethylmaleimide (NEM) (Sigma-Aldrich, E3876) to prevent post-lysis disulfide bond formation of sulfhydryls. The construct without a peptide insert (Δ ins) was used as a negative control, and a construct (mHC#2) from the original study[45] was used as a positive control. In Grp170 binding experiments, BiP was co-transfected to increase the expression of clients.

**Immune reagent sources.** The rabbit polyclonal anti-GRP170 (1:1000 for western blotting), and mouse monoclonal anti-ERdj4 (0.8ug/ml for western blotting) antibodies were produced in the Hendershot lab. The mouse monoclonal anti-HA

antibody was a kind gift from Dr. Al Reynolds (Vanderbilt University, 1:500 for IP, 1:1000 for western blotting). Other antibodies used were obtained commercially and include: goat anti-mouse λ LC (SouthernBiotech, 1060-01, 1:500 for IP,1:1000 for western blotting), HRP-conjugated goat anti-rabbit (Santa Cruz Biotechnology, sc-2054, 1:10,000), HRP-conjugated donkey anti-goat (Santa Cruz Biotechnology, sc-2020, 1:10,000), HRP-conjugated goat anti-mouse (SouthernBiotech, 1038-05, 1:10,000), monoclonal anti-ERdj5 (Abnova, H00054431-M01, 1:1,000 for western blotting), polyclonal rabbit anti-pro-SP-C (Seven Hills Bioreagents, WRAB9337, rabbit bleed 364, 1:2,000 for IP, 1:5000 for western blotting), mouse monoclonal anti-β-Actin (Sigma-Aldrich, AC-15, 1:2,000 for western blotting), and IRdye® secondary antibodies (LI-COR Biosciences, all 1:20,000): goat anti-mouse IgG (925-32210), goat anti-rabbit IgG (925-68071), and donkey anti-goat IgG (926-68074).

**Immunoprecipitation and quantitative western blotting.** Cell lysates were precleared by incubating with 30 μl of Protein A agarose slurry (CaptivA™ PriMAB, CA-PRI-0100) for 1 h at 4 °C under gentle rotation to reduce non-specific binding. Clarified lysates were transferred to new tubes for immunoprecipitation (IP) overnight at 4 °C under gentle rotation, and a portion of each lysate was removed to serve as an input control where indicated. Full-length SP-C proteins were isolated with anti-SP-C antibody, and in the case of SP-C peptide binding studies, samples were immunoprecipitated with goat anti-λ overnight, and in both cases Protein A agarose was added to samples the following morning and incubated at 4 °C for 1 h with gentle rotation. Immunoprecipitated material was washed 3 times with NP40 washing buffer (NP40 lysis buffer containing 0.4 M NaCl), heated for 5 min in either reducing or non-reducing Laemmli buffer, and separated by SDS-PAGE. After electrophoresis proteins were transferred to a PVDF membrane (Millipore, IPFL00010), specific proteins were detected by incubation with the indicated primary antibodies followed by appropriate species-specific secondary antibodies. In the case of HRP-conjugated secondary reagents, proteins were visualized by incubating the membranes with Pierce ECL western blotting substrate (ThermoFisher Scientific, 32106) and exposed to x-ray film or the LI-COR Odyssey Fc Imager (LI-COR Biosciences). When secondary antibodies conjugated to IRDye680 or IRDye800 (Li-COR) were employed, protein bands were detected using the LI-COR Odyssey CLx Imager (LI-COR Biosciences). In the case of co-chaperone binding to full-length SP-C L188Q and variants, western blotting to detect SP-C was performed using the Quick Western Blot kit (LI-COR, 926-69100) according to the manufacturer's instruction. The kit was used to distinguish pro-SP-C (~25 kDa) from the precipitating antibody, which also runs ~25 kDa. The precipitating antibody alone was added to one lane to control for residual signal. Images were acquired and analyzed using the Image Studio software (LI-COR).

**Radiolabeling.** Cells were plated in p60 dishes coated with poly-L-lysine. Twenty-four hours after transfection, cells were starved in 2 mL labeling media consisting of 1x DMEM without Cys and Met (Cys⁻/Met⁻) (Corning, 17-204CI) supplemented with 10% FBS dialyzed against PBS, 1% L-glutamine, and 1% antibiotic/antimycotic at 37 °C for 30 min and then labeled with Express [³⁵S] Labeling Mix (PerkinElmer, NEG072-007) for 30 min in the case of monitoring glycosylation status and in trans folding assays or labeled for 3-5 hr, as indicated when the oxidative state of the protein was being established. Cells were harvested and lysed with an appropriate lysis buffer as described for western blotting, and samples were processed for immunoprecipitation with the indicated antibodies. Gels were stained with Coomassie blue and de-stained prior to drying. Dried gels were exposed to BAS storage phosphor screens (Cytiva, 28956475) and scanned using a phosphor imager (Typhoon FLA 9500 GE Healthcare) to detect signals. Alternatively, to improve signals, gels containing labeled, immunoprecipitated proteins were transferred to a PVDF membrane instead of drying before exposing to phosphor screens. Signals were analyzed with the ImageQuant TL software (Cytiva).

**Endoglycosidase H treatment.** To determine the extent of membrane integration of the wild-type and L188Q mutant proteins, single N-linked glycosylation acceptor sites were engineered at defined locations. Transfected cells expressing these constructs were metabolically labeled and SP-C proteins were isolated by immunoprecipitation. Samples were split and treated with endoglycosidase H (Endo H) or Endo Hf (NEB, P0702, P0703) to remove glycans or incubated without Endo H as a control, according to the manufacturer's instructions. Samples were run on SDS-PAGE and processed for phosphorimaging.

**Quantitation.** In the case of co-chaperones binding to full-length SP-C proteins, one additional dish was co-transfected with cDNA encoding the co-chaperone and an empty vector (EV) without SP-C cDNA and immunoprecipitated with anti-SP-C to serve as a control for non-specific co-chaperone binding. The signals for the various SP-C proteins and the co-chaperone were determined for each sample. Any signal for co-chaperone binding to the control lane was first subtracted from each of the other co-chaperone signals, and then the co-chaperone signal was normalized to the SP-C protein signal. Finally, the amount of co-chaperone binding to wild-type or parental SP-C construct was set to 100, and the signal for co-chaperone binding to each of the other constructs was expressed as a percent of wild-type or parental. The average percentages were computed from three or more

independent experiments. Standard deviation (SD) and standard error of the mean (SEM) were determined. Error bars in graphical data represent SEM. GraphPad Prism 9.0 software was used for statistical analysis. The difference in co-chaperone binding between SP-C L188Q (parental) and each SP-C mutant (SP-C L188Q site 2, site 3, and site 2/3) was assessed using one-sample $t$ test. Differences between the binding of co-chaperones (GRP170 vs ERdj5, ERdj5 vs ERdj4, and GRP170 vs ERdj4) to each SP-C mutant were estimated using unpaired $t$ test. A threshold $p < 0.05$ was statistically significant.

To examine the ability of ILD-associated SP-C mutants to disrupt maturation of wild-type SP-C, a 3HA-tagged version of the L188Q mutant protein was co-expressed with untagged wild-type SP-C. Cells were labeled for 30 min in order to capture earlier intermediates in SP-C biosynthesis. Proteins were isolated from clarified lysates by immunoprecipitation with anti-HA or anti-SPC as indicated, and isolated proteins were divided and electrophoresed under reducing and non-reducing conditions. The samples analyzed under reducing conditions were used to assess the protein's ability to pass ER quality control, traffic to the Golgi, and acquire the addition of palmitic acid on two cysteines located at the N-terminus. The slower and faster migrating bands corresponding to palmitoylated and unmodified wild-type SPC, respectively, were quantified to determine palmitoylation percentage, as a measure of successful maturation of the protein.

The other half of the samples, which were analyzed under non-reducing conditions, were used to determine if co-expression of the HA-tagged L188Q mutant trapped the wild-type protein in disulfide-linked oligomers. Because the presence of untagged wild-type protein in the large oligomeric species could not be distinguished from the L188Q protein, we instead tested for a reduction in the monomeric species present under non-reducing conditions, which was readily distinguished from the mutant protein.

**Bioinformatic analysis**. Sequences were run using the TANGO algorithm[20] to predict aggregation-prone sites, and mutations were chosen to decrease predicted aggregation propensities. In the case of site 2, three amino acids were altered to lower the aggregation propensity, whereas a single amino acid substitution in site 3 was sufficient to reduce propensity to aggregate. Since Leu188 is included in site 3, and that L188Q is a disease-associated mutation, we also made constructs with L188Q. Site 1 peptide included the part of the TM that enters the ER lumen, and two sets of mutations were done to decrease aggregation propensities.

**Structural modeling**. Figures of BRICHOS domain (PDB: 2YAD) were prepared with PyMOL Molecular Graphics System (Schrödinger, LLC.).

**Reporting summary**. Further information on research design is available in the Nature Research Reporting Summary linked to this article.

## Data availability
Primary data for this study are included in the Source Data file provided with this paper. Source data are provided with this paper.

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

## Acknowledgements

The authors wish to thank Dr. Timothy Weaver at Cincinnati Children's Hospital Medical Center for providing DNA plasmids of pro-SP-C and advice on isolating the various proteins. We are grateful for technical assistance and valuable discussions provided by Mr. Christian Melendez-Suchi and Ms. Mary Carson Irvine. This work was supported by the American Lebanese Syrian Associated Charities of St. Jude Children's Research Hospital and by NIH Grant R01 GM54068 to L.M.H. The content is solely the responsibility of the authors and does not necessarily represent the official views of the National Institutes of Health.

## Author contributions

Study was developed by L.M.H. and K.F.R.P.-P., executed by K.F.R.P.-P., M.J.M., and A.R.F., written by L.M.H. and K.F.R.P.-P., and edited by L.M.H., K.F.R.P.-P., M.J.M., and A.R.F.

## Competing interests

The authors declare no competing interests.
