## [Peer Review File · Nature Communications]

Mapping SP-C co-chaperone binding sites reveals molecular consequences of disease-causing mutations on protein maturationREVIEWER COMMENTS

Reviewer #1 (Remarks to the Author):

In the manuscript entitled "Mapping ERdj4, ERdj5, and Grp170 binding sequences provides new insights into surfactant protein C mutations" Pobre-Piza et. al. aim to study the details of the interaction between the surfactant protein C (SP-C) and mutants of SP-C associated with Interstitial Lung Disease (ILD) and BiP co-chaperones ERdj4, ERdj5 and Grp170. The authors had previously determined that ERdj4, ERdj5 and Grp170 interacted with substrates with binding patterns distinct of those for BiP and ERdj3.

Using clever and well-designed experiments, the authors found that SP-C mutants (already known to bind ERdj4 and ERdj5) also bound to Grp170 and characterized the effect of the L188Q mutation on the BRICHOS domain. They showed that the mutation interferes with the formation of one of the two disulfide bonds, exposing aggregation prone sequences that could then be bound by the co-chaperones and causing the formation of large oligomers.

The authors found that ERdj4, ERdj5 and Grp180 bind to only one of the two cytosolic aggregation-prone sequences of the L188Q substrate identified via the TANGO algorithm, and showed how the different co-chaperones differentially bind to this region in the context of the full-length substrate.

The authors then set on the study of the mutants of SP-C and showed that in the cell, the wild-type protein is able to interact with the mutants, and that the mutant protein prevents the transition of the WT protein into the Golgi compartment. It is also postulated that the chaperoning function of the BRICHOS domain on the transmembrane region of the SP-C occurs in TRANS.

This study has elegant and nicely performed experiments. The results are overall solid, and the conclusions are supported by the data. However, the way the manuscript is organized poses a challenge to the reader. From the abstract and the Introduction, it seems like the main goal of the work is to extend the findings of their previous publication and determine if the TANGO algorithm predicts sequences in proteins that correspond to those sequences bound by ERdj4, ERdj5 and Grp170, only using SP-C as a tool to test this. Even though this point was addressed, the paper then diverges from this aim and set to explore the effect of the mutant proteins on the folding of the WT counterpart in the cell, etc. etc.

To me, this work has two main parts: One regarding the interaction between SP-C proteins and ERdj4, ERdj5 and Grp170, and how the binding of these chaperones to a substrate can be predicted using the

TANGO algorithm. The other one, is the structural study of the mutant proteins, the effect of the mutations on the SP-C structure, and the effect of the mutant proteins on trafficking of the wild type protein. Both parts of the study yield compelling and useful results, but from the way they are presented is very difficult to know up front what are the questions that the experiments answer. To me, the manuscript needs an up front, clear statement of the aims of the work and the questions answered.

Other points:

1) The TANGO algorithm identifies aggregation-prone sequences. The way you state this gives me the idea that it ONLY identifies those sequences that have the features of ERdj4, ERdj5 or Grp170 binding sequences. I assume TANGO would predict some sequences also bound by Hsp70s and other DnaJ proteins. Did I misinterpret this? It is somehow confusing...

2) The introduction ends in a blunt way, without leading the reader into what will be answered. AS I mentioned before, from the last paragraph of the introduction I take that the aim of the project is to correlate the sequences predicted by TANGO (as B-structure aggregation-prone) with binding to ERdj4, ERdj5 and Grp170.

3) The Title of the result section “The BRICHOS domain is stabilized by two disulfide bonds, one of which is the target of multiple disease-associated mutations” is misleading. This section indeed shows that the disulfide bond is compromised in the mutant, but the fact that the domain has two disulfide bonds is not a result from this section...

4) The experiments described in the result section “binding of co-chaperones to peptides....” (pag 6) are very elegant. Could you please give a very brief description of the experimental strategy, so the reader can understand the experiment while reading? Something like described in the figure legend for Figure S4. Please make the membrane in Figure S4 darker (gray) as it is very difficult to see.

5) Only the first paragraph of the discussion talks about the (lack of) “correlation” between the aggregation-prone sequences predicted by TANGO and the binding to the ERdj4, 5 and Grp170. All the other 4 pages of discussion are dedicated to discuss the results obtained for the folding of the SP-C protein and the mutants. This is why I am surprised that the aim of the manuscript in all sections before seems to be focused on the co-chaperones binding.

6) The discussion contains an unnecessary description of the results.

7) Figure 1 legend. Cells were transfected with “DNA encoding the indicated co-chaperones” (not with the co-chaperones) or “transfected to express....”

8) Figure 2A. Could you please add to the structure of SP-C the part of the protein that constitutes the “mature” peptide (after removal of the BRICHOS domain)? The numbers for the Y axis in Fig 2B are very small.

9) Figure 3B. The figure does not describe what I am seeing (western blot? SDS-PAGE? SDS-PAGE of Radio-labeled proteins? You say “metabolically labeled” but that could mean many things. You could help the reader understand the figure without having to go to Materials and methods.

10) Figure 5. For the quantification of the bands in panel A, did you normalize the bands’ signals with the signals coming from the a-SP-C blot? For example, were the bands in the top panel (blotted with a-GRP170) normalized for the amount of protein in the blot using a-SP-C? The amount of protein for “site 2 mut” and “site 2/3 mut” seems lower than for “L188Q” and “site 3 mut”. That normalization would change the quantification results.

11) Figure S2A is unnecessary (same as Fig 3A)

12) Fig3 legend. Authors forgot to update the numbers for the percentages.

Reviewer #2 (Remarks to the Author):

The manuscript by Pobre-Piza et al. reports the interaction of GRP170, ERdj4 and ERdj5 with SP-C mutants associated with ILD, which is an important finding. TANGO prediction partially supports the ex vivo findings. Despite the previous analysis of two clients, it is vital to analyze multiple clients in this study to show the effectiveness of using TANGO algorithm to broadly predict the binding of GRP170, ERdj4 and ERdj5. Another main aspect that remains unclear is the consequence(s) of these three co-chaperone associations with the mutants of SP-C in cells. How do they individually or/and together target the SP-C mutants to ERQC? A detailed analysis of this aspect will add much value to the current work and will considerably advance the field. The reviewer understands that addressing the above

aspects requires considerable additional work. The reviewer supports reconsideration of the manuscript with major revisions.

Major issues:

1. Line 127: The reviewer requests that L188Q, C148S and C189S mutants of SP-C should be analyzed (as in Figure S3) in the same gel in order to conclusively support the statements in lines 127-131.

“..and very little was present as disulfide-bonded oligomers (Fig S3B)” unclear statement. Are the oligomers refer to aggregated species appearing as a smear under NR conditions or the band indicated by the “star” sign? What evidence the authors have to suggest that the smear does not contain similar oligomers/aggregates?

2. The authors state that “In this study we sought to build on this previous work and determine if TANGO algorithm can be used more broadly to predict binding patterns of BiP co-chaperones to additional clients” which appears to be one of the main objectives. Only site 2 is clearly demonstrated to bind the co-chaperones. This result and the fact only a single client is analyzed in this manuscript considerably weakens the outcome of the study and questions whether the TANGO algorithm can be used to broadly and effectively predict binding patterns of GRP170, ERdj4 and ERdj5 to client proteins. In order to claim that this work considerably advances previous findings, it is essential that the authors analyze a broad range of multiple clients.

3. Figure 5. Quantification of % binding of co-chaperones relative to the parental construct is not appropriate given that the expression of site 2 and site2/3 mutants have a much lower stability/ expression. The band intensities of the co-chaperones must be normalized to the respective pulldown levels of the SP-C.

4. Data presented in Figures 1, 2, and 3 can be easily combined to make Figure 1. Similarly, data in Figures 4 and 5 should be combined to one figure given that the findings are redundant and supporting the same conclusion.

5. Figure S1 is missing.

6. The data in Figure 6 suggest that the TM domain integration may also be somewhat defective in the L188Q mutant, possibly due to the overall misfolding of the protein, which is driven by the collapse of

the BRICHOS domain. What significance does this finding have on the overall story? Perhaps Figure 6 should be shifted to supplementary data to support other findings?

7. Line 226: “To examine the effects of mutant SP-C expression on oxidative folding of the wild-type protein SP-C and its transport to the Golgi, the L188Q mutant was modified with a 3X-HA peptide tag at its N-terminus...” Others have shown that (e.g. Lawson et al 2011 PNAS) mutant L188Q localizes to the ER and not Golgi. Here authors show that the ectopically expressed mutant L188Q still “interacts” or “co-aggregate” with the WT SP-C in cells, which is not surprising given that SP-C forms homomeric assemblies and misfolded versions e.g. Δexon4 can trap the WT proteins. The finding supports the previously established mechanism.

Minor issues:

Line 57: In vivo should be referred to as ex vivo if the experiments are done in cultured mammalian cells.

Line 83: The title is unclear.

Figure 1: Can the authors explain the reason for the observed (consistently) shift in SP-C L188Q vs the WT protein? Also, what is the band appearing around 30 kDa in lysates expressing WT protein and not the SP-C L188Q mutant? Could these differences arise from changes in PTMs and peptide cleavage? If so, can that also have an influence on co-chaperone and Bip binding?

Figure 1: Steady state expression levels of ERdj4 appear to vary considerably e.g. increased ERdj4 levels appear in lysates from SP-C L188Q and EV transfections. Please explain these discrepancies.

Line 94: Define beta aggregates.

Line 95: Indicate what the two clients are.

Line 116: Related to the question associated with Figure 1. The band representing unmodified SP-C L188Q under reduced conditions clearly runs higher to that of the WT protein under similar conditions. A better characterization of what this species represent is needed. Further, please comment on why the steady state levels of the WT protein are consistently lower than the mutant, which other studies have also reported – is this related to protein aggregation related turnover defects? (compare Fig 1 and Fig 3).

The % of the bands reported for L188Q NR in Fig. 3B: Do the authors see the same pattern in all the repeats? Perhaps repeat n = 3 or minimum include the repeat in the supplement since the experiment was done only two times?

Line 124: The authors may likely be right here, but it is difficult to interpret this aspect due to the relatively slow migration of the unmodified SP-C L188Q (reduced) compared to the WT protein (reduced) (See previous comments).

Line 26: "Oxidative misfolding" Please define clearly if such phrases are used. Are the authors suggesting that the misfolding and aggregation is driven by aberrant disulfide bond formations within the SP-C mutant – inter or intra? Can this be observed also in the Δ exon4 mutant?

Line 130: The statement regarding ILD-associated mutations is unclear.

Line 138: "in vivo" or "ex vivo"? If the experiments are conducted in cultured human cells, the correct term should be "ex vivo."

Line 138: In brief, explain how the ex vivo screen was performed for the mutants.

Reviewer #3 (Remarks to the Author):

The manuscript describes the characterization of aggregation-prone sites in surfactant protein C (SP-C) and their binding to Grp170, ERdj4, and ERdj5, all co-chaperones of the ER-resident Hsp70 BiP.

The study is original in that these sites are atypical binding sites for chaperones, as most (co)chaperone binding sites are buried in folded proteins but not necessarily aggregation prone. This report is an important addition to our understanding of chaperone binding, as well as to the functioning of chaperones in various processes.

The data also demonstrate negative effects on membrane integration of a C-terminal domain when the N-terminal domain is not folding properly. Ectodomain mutants of most single-pass type-I membrane proteins have a different phenotype: that of misfolding the ectodomain but with proper membrane integration.

The manuscript explains the autosomal dominant nature of ILD, through oligomerization of mutant SP-C with properly disulfide-bonded wild-type SP-C, leading to increase in oligomer size and retention of the wild-type protein in the ER.

The conclusions are supported by the data, but clarity could be improved a lot. The discussion reads well, is thorough and educational.

Specific comments:

Some things are not clear, probably because of unclear annotation, but sometimes because explanations are missing. Specifically:

Figure S3B, the C148S mutant: it looks like it does not get palmitoylated. Does it not get transported to the Golgi?

I do not understand line 130-131 on ILD, in relation to the data on C148S.

Explanation follows only in the discussion at line ~322

Lines 155 and 210: why not shown? Would be better to show.

And line 333 sounds like a good addition to the data as it would add to the events SP-C undergoes during its biosynthesis.

Figure 5 shows is convincing, and shows also that the site-2 mutation lowers expression of SP-C. This suggests that the aggregation-prone-binding sites are required for proper folding of SP-C.

Figure S7A is the quantitation of panel B? This is not a clear figure.

The legend mentions a second B (not bold). "Calculation....". Again not clear.

Minor comments:

Figure S1 is missing in text and figure list.

Line 125: please add reference for this, or add data.

Legend Figure S3 is incomplete, contains xx instead of numbers.

It helps to explain the color coding again in panel A

Please explain the cartoons in Figure S4, panels C and F

Clarity would improve when the annotation of Figure 5 would show that all mutants also have L188Q. And that these are in cis on the same cDNA. That is, if I understand properly.

Response to the reviewers

We thank the reviewers for their overall positive assessment of our study. However, there was a general consensus that the two parts of our study were not well-integrated and that a clearer description in the results section of the methods used would significantly increase the reader's ability to follow the results. To remedy this, we have more fully discussed how our attempts to identify co-chaperone binding sites and their usage in a full-length protein required an understanding of how disease-associated mutations in SP-C affected its folding, membrane integration, and maturation. We have also increased our description of the experimental methods in the text and figure legends, while maintaining very detailed descriptions in the Methods section. To address other points, we have combined our previous Figure 2 and supplemental Figure S2 and renamed this Figure S1, which reduced our main figures to 6. We significantly extended our discussion of insights learned on co-chaperone binding sequences, differences in the effects of our introduced mutation on the binding of individual co-chaperones to full-length SP-C as opposed to peptide, and limitations in the ability of the TANGO algorithm in identifying sites. To provide space for this, we removed excessive description of results and shortened the discussion of insights we obtained for the folding of the SP-C protein and the mutants. These changes in the revised manuscript are highlighted in gray. Lastly, we respond to each point raised by all three reviewers individually below.

Reviewer #1 (Remarks to the Author):

In the manuscript entitled "Mapping ERdj4, ERdj5, and Grp170 binding sequences provides new insights into surfactant protein C mutations" Pobre-Piza et. al. aim to study the details of the interaction between the surfactant protein C (SP-C) and mutants of SP-C associated with Interstitial Lung Disease (ILD) and BiP co-chaperones ERdj4, ERdj5 and Grp170. The authors had previously determined that ERdj4, ERdj5 and Grp170 interacted with substrates with binding patterns distinct of those for BiP and ERdj3.

Using clever and well-designed experiments, the authors found that SP-C mutants (already known to bind ERdj4 and ERdj5) also bound to Grp170 and characterized the effect of the L188Q mutation on the BRICHOS domain. They showed that the mutation interferes with the formation of one of the two disulfide bonds, exposing aggregation prone sequences that could then be bound by the co-chaperones and causing the formation of large oligomers.

The authors found that ERdj4, ERdj5 and Grp180 bind to only one of the two cytosolic aggregation-prone sequences of the L188Q substrate identified via the TANGO algorithm, and showed how the different co-chaperones differentially bind to this region in the context of the full-length substrate.

The authors then set on the study of the mutants of SP-C and showed that in the cell, the wild-type protein is able to interact with the mutants, and that the mutant protein prevents the transition of the WT protein into the Golgi compartment. It is also postulated that the chaperoning function of the BRICHOS domain on the transmembrane region of the SP-C occurs in TRANS.

This study has elegant and nicely performed experiments. The results are overall solid, and the conclusions are supported by the data. However, the way the manuscript is organized poses a challenge

to the reader. From the abstract and the Introduction, it seems like the main goal of the work is to extend the findings of their previous publication and determine if the TANGO algorithm predicts sequences in proteins that correspond to those sequences bound by Erdj4, ERdj5 and Grp170, only using SP-C as a tool to test this. Even though this point was addressed, the paper then diverges from this aim and set to explore the effect of the mutant proteins on the folding of the WT counterpart in the cell, etc. etc.

To me, this work has two main parts: One regarding the interaction between SP-C proteins and Erdj4, ERdj5 and Grp170, and how the binding of these chaperones to a substrate can be predicted using the TANGO algorithm. The other one, is the structural study of the mutant proteins, the effect of the mutations on the SP-C structure, and the effect of the mutant proteins on trafficking of the wild type protein. Both parts of the study yield compelling and useful results, but from the way they are presented is very difficult to know up front what are the questions that the experiments answer. To me, the manuscript needs an up front, clear statement of the aims of the work and the questions answered. *This reviewer is correct about the evolution of the study. We have modified the introduction and the abstract to better reflect this and explain how these two parts are closely tied.*

Other points:

1) The TANGO algorithm identifies aggregation-prone sequences. The way you state this gives me the idea that it ONLY identifies those sequences that have the features of ERdj4, ERdj5 or Grp170 binding sequences. I assume TANGO would predict some sequences also bound by Hsp70s and other DnaJ proteins. Did I misinterpret this? It is somehow confusing... *Yes, indeed these sequences are often identified by BiP and ERdj3 as reported in the previous paper published in Mol. Cell. We found that mutations that reduced TANGO's prediction of aggregation propensity and reduced co-chaperone binding, actually increased the binding of BiP to the peptide sequences and to the full-length protein. This might imply a competition in binding to these regions, and changes that disfavor co-chaperone binding are well-tolerated by BiP. We have tried to make this clearer in the description of the TANGO algorithm.*

2) The introduction ends in a blunt way, without leading the reader into what will be answered. AS I mentioned before, from the last paragraph of the introduction I take that the aim of the project is to correlate the sequences predicted by TANGO (as B-structure aggregation-prone) with binding to ERdj4, ERdj5 and Grp170. *Agreed. In our eagerness to keep the word count to the minimum, we clearly erred on this point. We have extended the introduction in the revised manuscript to address this and explain how our goal of examining these sequences in the context of a full-length protein, as opposed to just peptides, led to experiments that provided insights into SP-C folding and even the autosomal dominant nature of the disease.*

3) The Title of the result section "The BRICHOS domain is stabilized by two disulfide bonds, one of which is the target of multiple disease-associated mutations" is misleading. This section indeed shows that the disulfide bond is compromised in the mutant, but the fact that the domain has two disulfide bonds is not a result from this section... *Thank you for bringing this to our attention. We have replaced Figure 3 in the revised manuscript to include the second ILD-associated mutant, and a mutant in which the C120-C148 bond is disrupted. Reduced and non-reduced versions of these SP-C proteins are presented together in*

the new Figure 2 in order to make the case at this point in the paper. We have also worked to make the description of the data and our conclusions from it clearer.

4) The experiments described in the result section “binding of co-chaperones to peptides....” (pag 6) are very elegant. Could you please give a very brief description of the experimental strategy, so the reader can understand the experiment while reading? Something like described in the figure legend for Figure S4. Please make the membrane in Figure S4 darker (gray) as it is very difficult to see. *We have added a description of the experimental strategy in the text and made the membrane darker in Figure S4 (now Figure S3), so it is more readily seen.*

5) Only the first paragraph of the discussion talks about the (lack of) “correlation” between the aggregation-prone sequences predicted by TANGO and the binding to the ERdj4, 5 and Grp170. All the other 4 pages of discussion are dedicated to discuss the results obtained for the folding of the SP-C protein and the mutants. This is why I am surprised that the aim of the manuscript in all sections before seems to be focused on the co-chaperones binding. *We did not conclude that there is a complete lack of correlation between TANGO predictions and binding of the co-chaperones. Our data suggests some differences in the magnitude of effects of mutations on binding in the context of a full-length protein, and that both the length of the peptide and the relative aggregation propensity are important for recognition. We did find that mutations in Site 1, which TANGO predicted it to no longer be aggregation-prone, reduced binding of the co-chaperones but did not entirely inhibit them. We clarified in the text that this piece of data indicates the ability of the TANGO algorithm to predict co-chaperone binding or loss of binding was not absolute. Our uneven treatment of co-chaperone binding and SP-C folding in the discussion is well-taken. To remedy this concern, we have taken two tacks. The first is to increase the discussion of new insights gained on their binding preferences, and the second is to reduce our discussion of SP-C folding. However, as understanding how the various sites that the TANGO algorithm identified might be assessable to the co-chaperones, it was critical to explore the effects of the mutations on proper SP-C folding. We have tried to better integrate these two parts of the manuscript.*

6) The discussion contains an unnecessary description of the results. *We have removed much of the discussion of the results in the revised manuscript.*

7) Figure 1 legend. Cells were transfected with “DNA encoding the indicated co-chaperones” (not with the co-chaperones) or “transfected to express....” *Thank you. That was a bit sloppy on our part and has been appropriately edited.*

8) Figure 2A. Could you please add to the structure of SP-C the part of the protein that constitutes the “mature” peptide (after removal of the BRICHOS domain)? The numbers for the Y axis in Fig 2B are very small. *We have added a bar over the color-coded domain schematic and an oval outline in the topology cartoon to indicate this and have included the sequence of the mature protein. We also increased the font size on the Y axis of Figure 2B, which is now Figure S1B.*

9) Figure 3B. The figure does not describe what I am seeing (western blot? SDS-PAGE? SDS-PAGE of Radio-labeled proteins? You say “metabolically labeled” but that could mean many things. You could help the reader understand the figure without having to go to Materials and methods. *This is now*

Figure 2B and 2C in the revised manuscript. We now indicate that these images are from radiolabeled proteins that have been immunoprecipitated and analyzed by either non-reducing (2B) or reducing (2C) SDS-PAGE.

10) Figure 5. For the quantification of the bands in panel A, did you normalize the bands' signals with the signals coming from the a-SP-C blot? For example, were the bands in the top panel (blotted with a-GRP170) normalized for the amount of protein in the blot using a-SP-C? The amount of protein for "site 2 mut" and "site 2/3 mut" seems lower than for "L188Q" and "site 3 mut". That normalization would change the quantification results. *Yes, this is a very important point. Because the levels of the various SP-C proteins were affected by alterations in co-chaperone binding, we did indeed normalize the signals. We had described this in the Materials and Methods but have made this clearer in the revised text of the results section and in the figure legend. This is now Figure 4 in the revised manuscript.*

11) Figure S2A is unnecessary (same as Fig 3A) *Agreed. Figure S2A and Figure 2 have now been combined and renamed Figure S1. The image used shows where sites 2 and 3 map to the ribbon structure. Figure 3A (now Figure 2) has been replaced with an image that better shows the location of the two disulfide bonds. This image is repeated in Figure S2A, as both figures included experiments to address the formation of those disulfide bonds, and we feel having it in both places will help the reader follow the experiments and interpretations better.*

12) FigS3 legend. Authors forgot to update the numbers for the percentages. *Oh dear, thank you for pointing this out. It has been corrected in the revised manuscript.*

Reviewer #2 (Remarks to the Author):

The manuscript by Pobre-Piza et al. reports the interaction of GRP170, ERdj4 and ERdj5 with SP-C mutants associated with ILD, which is an important finding. TANGO prediction partially supports the ex vivo findings. Despite the previous analysis of two clients, it is vital to analyze multiple clients in this study to show the effectiveness of using TANGO algorithm to broadly predict the binding of GRP170, ERdj4 and ERdj5. Another main aspect that remains unclear is the consequence(s) of these three co-chaperone associations with the mutants of SP-C in cells. How do they individually or/and together target the SP-C mutants to ERQC? A detailed analysis of this aspect will add much value to the current work and will considerably advance the field. The reviewer understands that addressing the above aspects requires considerable additional work. The reviewer supports reconsideration of the manuscript with major revisions. *We definitely agree that the next step is to identify the consequences of co-chaperone binding to these sequences, their order of interaction, and the requirements for the additional activities that two of these co-chaperones possess. While we are pursuing these questions, this represents an entire study in itself.*

Major issues:

1. Line 127: The reviewer requests that L188Q, C148S and C189S mutants of SP-C should be analyzed (as in Figure S3) in the same gel in order to conclusively support the statements in lines 127-131. *We have*

now included the wild-type protein, the mutants, as well as the Δ exon 4 protein on the same reducing and non-reducing gels in the revised manuscript (Figure 2B and 2C). We have retained Figure S3 (now Figure S2B) in order to analyze reduced and non-reduced samples of individual proteins together on the same gel. This allowed us to observe changes in mobility of monomers due to longer range intramolecular disulfide bond formation. Because these shifts in mobility are rather small, we find it is important to have them next to each other. However, the diffusion of 2-ME as the gels are running necessitates that the reduced and non-reduced samples be separated by at least two lanes. Thus for this point, the constructs could not all be analyzed on the same gel.

*“..and very little was present as disulfide-bonded oligomers (Fig S3B)” unclear statement. Are the oligomers refer to aggregated species appearing as a smear under NR conditions or the band indicated by the “star” sign? What evidence the authors have to suggest that the smear does not contain similar oligomers/aggregates? *Initially we were only including the material at the very top of the gels, as there was very little signal below this for the L188Q mutant. However, by including more mutants, two different labeling times, and enhancing our method of detection in the more recently obtained data, we indeed observe a smear in some cases that is not present in the reduced samples. We now include this entire area in our analysis of disulfide-linked oligomers and subtract any background signal in the non-transfected lanes in this region of the non-reducing gel. This is clarified in the revised materials and methods and text.**

2. The authors state that “In this study we sought to build on this previous work and determine if TANGO algorithm can be used more broadly to predict binding patterns of BiP co-chaperones to additional clients” which appears to be one of the main objectives. Only site 2 is clearly demonstrated to bind the co-chaperones. This result and the fact only a single client is analyzed in this manuscript considerably weakens the outcome of the study and questions whether the TANGO algorithm can be used to broadly and effectively predict binding patterns of GRP170, ERdj4 and ERdj5 to client proteins. In order to claim that this work considerably advances previous findings, it is essential that the authors analyze a broad range of multiple clients. *At issue here is that very few proteins have as yet been identified as clients of either GRP170 or ERdj4. ERdj5 is a PDI family member with reductase activity and thus interacts directly with disulfide bonds as well. The fact that mutation of site 2 in the full-length protein had a greater effect on GRP170 and ERdj4 binding is consistent with this. However, we take issue with the comment that only a single client is added. In the original study 5 peptides were identified by the TANGO algorithm, which all bound the three co-chaperones, and we have added two more. Simple mutations that reduce TANGO-predicted aggregation propensity disrupt binding of all three co-chaperones to one of them, but we find for the first time that mutations which completely altered TANGO-predicted aggregation did not completely inhibit co-chaperone binding. The current study also identifies difference effects of mutations on the magnitude of co-chaperone binding in the full-length protein that are not observed in the peptides indicating context-dependent effects on binding of the various co-chaperone. Finally, our studies indicate that a combination of predicted aggregation propensity and size of the aggregation prone sequence are important parameters for co-chaperone binding. We have extended our discussion of these points.*

3. Figure 5. Quantification of % binding of co-chaperones relative to the parental construct is not appropriate given that the expression of site 2 and site2/3 mutants have a much lower stability/

expression. The band intensities of the co-chaperones must be normalized to the respective pulldown levels of the SP-C. *We completely agree this is an important point and had normalized for client levels, as described in the materials and methods section. We should have taken more care on emphasizing this point and now make this clearer in the manuscript text and figure legend as well.*

4. Data presented in Figures 1, 2, and 3 can be easily combined to make Figure 1. Similarly, data in Figures 4 and 5 should be combined to one figure given that the findings are redundant and supporting the same conclusion. *We disagree and think that Figures 1, 2 and 3 make very different points. Figure 1 demonstrates that GRP170 also binds to SP-C mutants, which is a critical point if all three co-chaperones recognize the same sequence. Figure 2 depicts where TANGO predicts aggregation-prone sequences exist and with supplemental Figure 2 reveals that two of them are largely buried in the folded BRICHOS domain. We have combined them in the revised manuscript as Figure 1S. Figure 3 (now Figure 2) demonstrates that mutation of a disease-associated point mutation in this domain has a dramatic effect on its correct oxidative folding, which could make these sites accessible to the chaperones, and further demonstrates that disruption of the second disulfide bond, which has not been targeted in ILD, has less of an effect on oxidative folding.*

5. Figure S1 is missing. *We did not include supplemental data for Figure 1. The numbering system we used was to indicate which figure the supplemental data corresponded to. We have combined Figure 2 and Figure S2, which are Figure 1S in the revised manuscript.*

6. The data in Figure 6 suggest that the TM domain integration may also be somewhat defective in the L188Q mutant, possibly due to the overall misfolding of the protein, which is driven by the collapse of the BRICHOS domain. What significance does this finding have on the overall story? Perhaps Figure 6 should be shifted to supplementary data to support other findings? *We believe this is a very critical component of the study. This data indicates that indeed TM domain integration is adversely affected by a mutant BRICHOS domain and that this allows an additional aggregation prone sequence to enter the ER where it could be bound by the co-chaperones studied here.*

7. Line 226: "To examine the effects of mutant SP-C expression on oxidative folding of the wild-type protein SP-C and its transport to the Golgi, the L188Q mutant was modified with a 3X-HA peptide tag at its N-terminus..." Others have shown that (e.g. Lawson et al 2011 PNAS) mutant L188Q localizes to the ER and not Golgi. Here authors show that the ectopically expressed mutant L188Q still "interacts" or "co-aggregate" with the WT SP-C in cells, which is not surprising given that SP-C forms homomeric assemblies and misfolded versions e.g. Δexon4 can trap the WT proteins. The finding supports the previously established mechanism. *We were aware of that study and had cited it. While this might be inferred, in fact there are no experimental data to indicate that aggregation prone SP-C mutants have a dominant negative effect on the WT protein. Our data show that this doesn't occur by a mutant BRICHOS domain failing to integrate the WT transmembrane domain in trans. Further it shows that a folded and unfolded SP-C protein can still interact, which has previously been reported to occur through the transmembrane domain interactions. Since the mutant TM does not integrate correctly and instead forms an aggregation prone β strand, it was not clear that assembly would still occur. Furthermore, our studies show that the consequence of this is that the chaperone-bound mutant exerts a dominant effect on the WT protein, which is now co-retained with the mutant protein. Thus, ILD is not simply due to*

haplo-insufficiency of native protein.

Minor issues:

Line 57: In vivo should be referred to as ex vivo if the experiments are done in cultured mammalian cells. *We appreciate this distinction and have changed in vivo to say either in cultured cells or ex vivo.*

Line 83: The title is unclear. *We have modified it better represent our findings.*

Figure 1: Can the authors explain the reason for the observed (consistently) shift in SP-C L188Q vs the WT protein? Also, what is the band appearing around 30 kDa in lysates expressing WT protein and not the SP-C L188Q mutant? Could these differences arise from changes in PTMs and peptide cleavage? If so, can that also have an influence on co-chaperone and Bip binding? *We do not know why the L188Q runs slightly slower on our gels, but the reviewer is correct in pointing out that the wild-type protein undergoes multiple proteolytic cleavages as it travels through post-ER compartments of the secretory pathway. This is something that does not happen to the L188Q mutant, although we cannot suggest this is the cause with any certainty. The band at ~30kDa in the WT lane represents palmitoylated SP-C. This PTM occurs in the Golgi, and because the L188Q and Δ exon 4 mutants are retained in the ER, they are not modified by palmitoylation. These points have been well-established by other investigators and have been cited in the text. The disruption in folding of the BRICHOS domain we observed in the mutants, leads to co-chaperone (and BiP) binding causing them to be retained in the ER.*

Figure 1: Steady state expression levels of ERdj4 appear to vary considerably e.g. increased ERdj4 levels appear in lysates from SP-C L188Q and EV transfections. Please explain these discrepancies. *We agree that there was a slight increase in ERdj4 levels in these two samples. This was not observed in subsequent experiments, so we have no reason to think that co-expression of WT or the delta exon 4 mutants reduce expression of ERdj4. The point of this experiment was to determine qualitatively if GRP170 also bound to this client. For all qualitative binding experiments, we normalized for any differences in expression of co-chaperones and clients.*

Line 94: Define beta aggregates. *The term beta aggregates refers to a type of protein aggregate arising from abnormal intermolecular associations of beta sheet regions. We define this term in the revised manuscript.*

Line 95: Indicate what the two clients are. *We apologize for not naming them again in the results. We now do so. The original study examined sequences from a non-secreted immunoglobulin light chain and the two domains of the Ig heavy chain that are the focus of ER quality control. Both proteins are ~200 amino acids in size and each is comprised of two Ig domains.*

Line 116: Related to the question associated with Figure 1. The band representing unmodified SP-C L188Q under reduced conditions clearly runs higher to that of the WT protein under similar conditions. A better characterization of what this species represent is needed. *This is similar to the point raised above. We do not know why it runs slower on our gels. We have sequenced it in entirety and there are no additional deviations from the wild-type sequence. It is not uncommon for non-conservative amino*

acid changes to affect mobility due to effects on detergent binding. We realize that this difference was not observed in the 2008 Weaver paper. We ran higher percent acrylamide gels that were run particularly long in an attempt to separate SP-C from ERdj4 and the immunoprecipitating antibody that all have similar molecular weights, which could contribute to our ability to detect differences in their mobility.

Further, please comment on why the steady state levels of the WT protein are consistently lower than the mutant, which other studies have also reported – is this related to protein aggregation related turnover defects? (compare Fig 1 and Fig 3). *While that might contribute to higher levels of the mutants, an additional factor is the fact that the wild-type protein traffics to the Golgi upon passing ER quality control, and its quickly subjected to proteolytic processing in this organelle. This is why some experiments were conducted with short labeling times, and in all cases the levels of the various SP-C proteins were normalized when co-chaperone binding was being measured.*

The % of the bands reported for L188Q NR in Fig. 3B: Do the authors see the same pattern in all the repeats? Perhaps repeat n = 3 or minimum include the repeat in the supplement since the experiment was done only two times? *We are very sorry for this error on our part. The figure legends were not updated to include the final numbers before submitting. All number of biological replicates are clearly indicated now. While most had at least three biological replicates (n=3), some had 5 or more.*

Line 124: The authors may likely be right here, but it is difficult to interpret this aspect due to the relatively slow migration of the unmodified SP-C L188Q (reduced) compared to the WT protein (reduced) (See previous comments). *The critical comparison is of the non-reduced to the reduced for each individual SP-C protein, which allows the formation of intramolecular disulfide bonds to be detected. We have attempted to make this clearer in the text.*

Line 26: “Oxidative misfolding” Please define clearly if such phrases are used. Are the authors suggesting that the misfolding and aggregation is driven by aberrant disulfide bond formations within the SP-C mutant – inter or intra? *Yes, this is what was meant. The bonds are almost entirely intermolecular, as there were very few oxidized monomers present in the gel, which could include incorrect intramolecular bonds as well, although we did not attempt to determine if this small population was a mixture. We have clarified this in the revised text.*

Can this be observed also in the Δ exon4 mutant? *We have included new data to address this point. The exon 4 deletion removes one of the cysteines involved in each of the two disulfide bonds (C120 and C121 respectively), so neither of the native bonds can form in this mutant. Instead, we found that a predominant species involves a non-native bond between C148 and C189, and less abundant populations of intermolecular bonded oligomers. There are fewer free cysteines available to form the large disulfide-linked oligomers observed in the L188Q mutant.*

Line 130: The statement regarding ILD-associated mutations is unclear. *There are no disease-associated mutations that alter either cysteine involved in this disulfide bond. Given the large number of mutations causing ILD that have been identified and the fact that 4 mutations directly alter the cysteines comprising the C121-C189 bond and three more change the adjacent L188 amino acid. We found that*

mutation of the C120-C148 bond had only minor effects on intermolecular disulfide bonds, and thus conclude that the lack of disease-associated mutations affecting this bond further supported our finding that this bond was less important to maintaining soluble monomers.

Line 138: “in vivo” or “ex vivo”? If the experiments are conducted in cultured human cells, the correct term should be “ex vivo.” *Agreed and we have changed this as stated above.*

Line 138: In brief, explain how the ex vivo screen was performed for the mutants. *As per a similar comment from reviewer #1, we have included this with greater detail in the revised manuscript.*

Reviewer #3 (Remarks to the Author):

The manuscript describes the characterization of aggregation-prone sites in surfactant protein C (SP-C) and their binding to Grp170, ERdj4, and ERdj5, all co-chaperones of the ER-resident Hsp70 BiP.

The study is original in that these sites are atypical binding sites for chaperones, as most (co)chaperone binding sites are buried in folded proteins but not necessarily aggregation prone. This report is an important addition to our understanding of chaperone binding, as well as to the functioning of chaperones in various processes.

The data also demonstrate negative effects on membrane integration of a C-terminal domain when the N-terminal domain is not folding properly. Ectodomain mutants of most single-pass type-I membrane proteins have a different phenotype: that of misfolding the ectodomain but with proper membrane integration.

The manuscript explains the autosomal dominant nature of ILD, through oligomerization of mutant SP-C with properly disulfide-bonded wild-type SP-C, leading to increase in oligomer size and retention of the wild-type protein in the ER.

The conclusions are supported by the data, but clarity could be improved a lot. The discussion reads well, is thorough and educational.

Specific comments:

Some things are not clear, probably because of unclear annotation, but sometimes because explanations are missing. Specifically:

Figure S3B, the C148S mutant: it looks like it does not get palmitoylated. Does it not get transported to the Golgi? *We have run additional experiments to examine this more closely. A portion of this mutant does get palmitoylated, but this is not as large of a percentage as is observed with the wild-type protein. It is possible that the remaining cysteine, if exposed, subjects this protein to thiol-mediated retention.*

I do not understand line 130-131 on ILD, in relation to the data on C148S.

Explanation follows only in the discussion at line ~322. *Reviewer #2 made the same point. We now make this clearer in the description of the data in the results.*

Lines 155 and 210: why not shown? Would be better to show. We attempted several times to get numbers for the increase in BiP binding to the full-length SP-C mutant when co-chaperone binding was inhibited, but the presence of a background band at the same size as some of the constructs made it difficult to normalize for the amount of construct present. We are providing the data here for the reviewer, but this caveat makes us feel it is better not to include it in the manuscript.

And line 333 sounds like a good addition to the data as it would add to the events SP-C undergoes during its biosynthesis. *This has been added to Figure S5A.*

Figure 5 shows is convincing, and shows also that the site-2 mutation lowers expression of SP-C. This suggests that the aggregation-prone-binding sites are required for proper folding of SP-C. *The reviewer makes an interesting point. Indeed, this entire stretch of the linker domain is essential. We tried to introduce several mutations in site 2 that would reduce its aggregation propensity, but all of them affected maturation of the wild-type protein when they were introduced there. It is noteworthy that this region is a hotspot for ILD-associated mutations.*

Figure S7A is the quantitation of panel B? This is not a clear figure. *Figure S7A, now Figure S6A is the quantitation of the indicated aspects of Figure 6. We have clarified this in the legend*
 The legend mentions a second B (not bold). "Calculation....". Again not clear. *Figure S6B represents the specificity controls for immunoprecipitation of the various constructs used in Figure 6. This has been clarified in the revised manuscript.*

Minor comments:

Figure S1 is missing in text and figure list. *Figure 1 did not have supplemental data. Our numbering system for supplemental data was designed to refer to which of the primary figures it corresponded to. In response to a comment by Reviewer #2, we have combined Figure 2 and supplemental Figure 2 into Figure S1 in the revised manuscript.*

Line 125: please add reference for this, or add data. *This comment refers to the effect of this mutation in combination with that of the mutation of the second bond which is discussed later in the manuscript. We now include the second mutant in this figure (now Figure 2) and discuss them together.*

Legend Figure S3 is incomplete, contains xx instead of numbers. *This was an unfortunate error on our part. The XX was a place holder, as we waited for the final numbers to be calculated. They went into the text for the results, but through an oversight on my part they did not go into the figure legend. This has been corrected.*

It helps to explain the color coding again in panel A. *This has been added to the figure legend.*

Please explain the cartoons in Figure S4, panels C and F. *This has been added to the figure legend.*

Clarity would improve when the annotation of Figure 5 would show that all mutants also have L188Q. And that these are in cis on the same cDNA. That is, if I understand properly. *It is. Thank you for the suggestion. We have attempt to make this clearer in the text and figure legend.*

REVIEWERS' COMMENTS

Reviewer #1 (Remarks to the Author):

The authors have addressed all the raised points. The manuscript has greatly improved and it is suitable for publication.

One point: At the end of the discussion, there is a summary of the "discoveries" made on this paper. Avoid using "confirm" to describe what it was done.

Reviewer #2 (Remarks to the Author):

The main text of the revised manuscript has improved. The discussion section still contains content that is largely redundant with results section and can be further trimmed. There are still a considerable number of sentences throughout the manuscript that have structural issues and need careful editing to remove ambiguity.

Major comments

- Line 148: "...which was likely the C120 and C148 bond, since the L188Q mutation appeared to dramatically affect formation of the native C121 – C189 bond." Lines 154: "very few were large species that poorly entered the gel. Additionally, the 60% of the protein that remained monomeric formed an intramolecular disulfide bond that likely represents the C121 – C189 bond (Figure S2B)." Aggregation of SP-C as a consequence of L188Q mutation is clear. However, it is not clear whether this is occurring due to the disruption of the C189 – C121 bond. It is extremely important to show that disruption of C189 – C121 bond by mutating C189S results in the same outcome as L188Q. To experimentally outline what is happening here and directly demonstrate the importance of C189 – C121 bond, key controls analyzing mutants e.g. L188Q+C148S, C189S, L188Q+C189S must be shown under NR and R states.

- The study shows ILD-associated mutations considerably aggregate. The mutants also show increased Grp170, ERdj4 and ERdj5 co-chaperone binding. How these two aspects are linked and the fate of the mutants as a consequence of the co-chaperone interactions remains unclear. This aspect needs to be clearly dissected to complete the study. Others have indeed demonstrated that over-expressing ERdj4 and ERdj5 results in accelerated degradation of the ILD SP-C mutants, but such over-expression studies

may not reflect the ground reality. To provide a better picture a) Report the half-life of the WT vs mutant SP-C variants (e.g. via pulse chase) b) Report the half-life of the WT vs mutant SP-C variants in Grp170, ERdj4 and ERdj5 knockdowns (can be done as single, double, & triple knockdowns), c) Monitor the aggregation level of the WT vs mutant SP-C variants in the knockdowns and in MG132 treated cells. Such relatively simple, quick and quite doable experiments in combination with what is known, will provide a clear picture of the cause and effect of these mutations in SP-C in ILD. The co-chaperone binding defective mutants can be used as controls.

In order to significantly improve the impact of the findings and make this study worthy of publishing in Nature Communications, experimentally addressing the above two key points is needed.

Minor comments

- Figure 1 simply shows increased association of the 3x co-chaps with ILD mutants compared to WT SP-C. This is insufficient to justify a full figure. Reviewer again highly recommends combining Fig 1 with Fig 2 as the findings go together.
- Fig 2A: Please clearly show/ label the disulfide bonds that are formed between C120 and C148 and C121 and C189 in the structure. At least show two views of the ribbon structure so that the reader can see clearly the side chains and the disulfide bonds they form. The disulfide bonds can be shown with dotted lines?
- Line 136: Radio labeled with what? Indicate the isotope. Please provide key information in the main text as well.
- Fig 2C. A minor species resulting from palmitoylation is indicated for C148S. However, this species is not detected under reduced conditions in Fig S2. Please either repeat these experiments and provide the most representative images to avoid confusion or explain.
- Line 201: "To determine if either of these sites was responsible for co-chaperone binding to the disease-associated SP-C proteins, we engineered the site 2 and 3 mutations onto the full-length L188Q protein both individually and together and tested the altered L188Q constructs for co-chaperone binding using immunoprecipitation-coupled western blot analyses (Figure 4A)."

The sentence should be improved. Please restructure to remove any ambiguity as to what "these sites" are.

Reviewer #3 (Remarks to the Author):

The manuscript text and figures have much improved and are more accessible.

Two specific comments:

Please add to Figure S2B the location of exon 4, and the numbers of the cysteines in larger font.

The response to my comment: "Lines 155 and 210: why not shown? Would be better to show."

I completely sympathize with the frequent impossibilities of quantitations. My suggestion is to add the gels, and write in the legend the numbers for the fold changes (~6x, 20x, ~13x) plus the caveat that quantitation is the site1 mutants is not trustworthy because of the background band. The data are convincing as shown.

My other comments have been addressed well.

Response to reviewers

Reviewer #1 (Remarks to the Author):

The authors have addressed all the raised points. The manuscript has greatly improved and it is suitable for publication. *Our thanks to this reviewer for helping to improve our manuscript*

One point: At the end of the discussion, there is a summary of the "discoveries" made on this paper. Avoid using "confirm" to describe what it was done. *We have rewritten the summary to focus on the conclusions of this paper, which no longer includes the word confirm.*

Reviewer #2 (Remarks to the Author):

The main text of the revised manuscript has improved. The discussion section still contains content that is largely redundant with results section and can be further trimmed. There are still a considerable number of sentences throughout the manuscript that have structural issues and need careful editing to remove ambiguity. *The manuscript has been carefully edited to ensure greater clarity of our findings and their discussion.*

Major comments

- Line 148: "...which was likely the C120 and C148 bond, since the L188Q mutation appeared to dramatically affect formation of the native C121 – C189 bond." Lines 154: "very few were large species that poorly entered the gel. Additionally, the 60% of the protein that remained monomeric formed an intramolecular disulfide bond that likely represents the C121 – C189 bond (Figure S2B)." Aggregation of SP-C as a consequence of L188Q mutation is clear. However, it is not clear whether this is occurring due to the disruption of the C189 – C121 bond. It is extremely important to show that disruption of C189 – C121 bond by mutating C189S results in the same outcome as L188Q. To experimentally outline what is happening here and directly demonstrate the importance of C189 – C121 bond, key controls analyzing mutants e.g. L188Q+C148S, C189S, L188Q+C189S must be shown under NR and R states. *We engineered the C189S mutant and found that, similar to the L188Q mutant, it forms large disulfide-linked oligomers and there is no evidence of oxidized monomers. This confirms that 1) the L188Q mutant disrupts the C121-C189 disulfide bond, and 2) that the formation of this bond is critical to the formation of the C120-C148 bond. This data analyzed under both reducing and non-reducing conditions is included in Figure 2 and Supplemental Figure 2. The other two constructs that were suggested to make on the L188Q background would be uninformative, since the amount of oxidized monomers that these mutant would hope to disrupt are less than 10%.*
- The study shows ILD-associated mutations considerably aggregate. The mutants also show increased Grp170, ERdj4 and ERdj5 co-chaperone binding. How these two aspects are linked and the fate of the mutants as a consequence of the co-chaperone interactions remains unclear. This aspect needs to be clearly dissected to complete the study. Others have indeed demonstrated that over-expressing ERdj4 and ERdj5 results in accelerated degradation of the ILD SP-C mutants, but such over-expression studies

may not reflect the ground reality. To provide a better picture a) Report the half-life of the WT vs mutant SP-C variants (e.g. via pulse chase) b) Report the half-life of the WT vs mutant SP-C variants in Grp170, ERdj4 and ERdj5 knockdowns (can be done as single, double, & triple knockdowns), c) Monitor the aggregation level of the WT vs mutant SP-C variants in the knockdowns and in MG132 treated cells. Such relatively simple, quick and quite doable experiments in combination with what is known, will provide a clear picture of the cause and effect of these mutations in SP-C in ILD. The co-chaperone binding defective mutants can be used as controls.

We agree this an interesting and important avenue of research but respectfully argue that it is beyond the scope of the present study.

In order to significantly improve the impact of the findings and make this study worthy of publishing in Nature Communications, experimentally addressing the above two key points is needed.

Minor comments

- Figure 1 simply shows increased association of the 3x co-chaps with ILD mutants compared to WT SP-C. This is insufficient to justify a full figure. Reviewer again highly recommends combining Fig 1 with Fig 2 as the findings go together. *We fail to see why the reviewer feels the two figures should be combined. Figure 1 demonstrates that all three co-chaperones bind SP-C mutants using ip-western assays. Figure 2 shows the effects of the mutations on oxidative folding of the constructs and includes additional mutants to assess the contribution of each disulfide bond to structural integrity using metabolic labeling and electrophoresis under reducing and non-reducing conditions.*

- Fig 2A: Please clearly show/ label the disulfide bonds that are formed between C120 and C148 and C121 and C189 in the structure. At least show two views of the ribbon structure so that the reader can see clearly the side chains and the disulfide bonds they form. The disulfide bonds can be shown with dotted lines? *Three views of the structure with disulfide bonds were included in the last submission. The structure provided in Figure 2a includes side-chains for the four cysteines and clearly depicts each of the bonds (C120-C148 in pink and C121-C189 in blue), as do Supplemental Figures 1c and 2a using the same coloring scheme.*

- Line 136: Radio labeled with what? Indicate the isotope. Please provide key information in the main text as well. *This is described in the methods section. The Express [³⁵S] Labeling Mix (PerkinElmer, NEG072-007) was used.*

- Fig 2C. A minor species resulting from palmitoylation is indicated for C148S. However, this species is not detected under reduced conditions in Fig S2. Please either repeat these experiments and provide the most representative images to avoid confusion or explain. *Figure S2b was not exposed as long as Figure 2C in order to detect mobility differences that are lost with longer exposures. The band is definitely there in S2b if one looks carefully.*

- Line 201: “To determine if either of these sites was responsible for co-chaperone binding to the disease-associated SP-C proteins, we engineered the site 2 and 3 mutations onto the full-length L188Q protein both individually and together and tested the altered L188Q constructs for co-chaperone binding using immunoprecipitation-coupled western blot analyses (Figure 4A).”

The sentence should be improved. Please restructure to remove any ambiguity as to what “these sites”

are. *We agree this was clumsy and have broken the sentence into two with fewer pronouns!*

Reviewer #3 (Remarks to the Author):

The manuscript text and figures have much improved and are more accessible. *We thank this reviewer*

Two specific comments:

Please add to Figure S2B the location of exon 4, and the numbers of the cysteines in larger font. *We have increased the font size for the cysteines. We assume the reviewer means Fig S1b, as Fig S2b are western blots. The boundaries of exon4 were indicated with a dashed line in Fig S1b, but we have now added the words "Δexon4 " and "L188Q" to this panel. We were unable to indicate the boundaries of exon 4 on the ribbon structure as it obliterated everything else. Exon4 is also indicated on the diagram in Fig S1a.*

The response to my comment: "Lines 155 and 210: why not shown? Would be better to show."

I completely sympathize with the frequent impossibilities of quantitations. My suggestion is to add the gels, and write in the legend the numbers for the fold changes (~6x, 20x, ~13x) plus the caveat that quantitation is the site1 mutants is not trustworthy because of the background band. The data are convincing as shown. *We thank the reviewer for this suggestion but decided instead to remove this sentence.*

My other comments have been addressed well.